# Advanced insights into biomass burning aerosols during the 2023 Canadian wildfires from dual-site Raman and fluorescence lidar observations

Qiaoyun Hu<sup>1</sup>, Philippe Goloub<sup>1</sup>, Igor Veselovskii<sup>2</sup>, Thierry Podvin<sup>1</sup>, Gaël Dubois<sup>1</sup>, Sergey Khaykin<sup>3</sup>, William Boissière<sup>1</sup>, Fabrice Ducos<sup>4</sup>, and Mikhail Korenskiy<sup>2</sup>

Abstract. This study presents lidar observations of long-range transported biomass burning aerosol (BBA) plumes from the exceptional 2023 Canadian wildfire season, recorded between May and September at the ATOLL observatory (France) and the GPI site (Russia). ATOLL operates a multi-wavelength Raman lidar with 3 polarization channels (355, 532 and 1064 nm) and a single fluorescence channel at 466 nm. GPI uses a fluorescence lidar with 5 broadband fluorescence channels excited by 355 nm. The dual-site dataset combines multi-wavelength elastic scattering and depolarization measurements with fluorescence observations, enabling a comprehensive characterization of BBA properties in the free troposphere (FT) and upper troposphere–lower stratosphere (UTLS). UTLS layers exhibit higher particle depolarization ratios, slightly lower lidar ratios, lower extinction- and backscatter-related Angström exponents, and a redshift in fluorescence spectral peaks. Cross-site comparisons show consistent fluorescence magnitudes and spectral shapes, highlighting the potential of coordinated multi-lidar fluorescence observations. Correlation analysis indicates that depolarization ratio, extinction-related Angström exponent, and fluorescence color ratio are moderately ( $r^2 \approx 0.61-0.68$ ) correlated with layer altitude, however, this correlation is not sufficient to confirm a solid altitude dependence. It is likely that altitude is an intermediate variable linked to other controlling factors such as injection height of the plume, in-layer temperature and the plume origin. In addition, we observed BBAs showing no clear hygroscopic growth at RH of 90%–100% and statistically low RH values in the detected nearly 100 layers, suggesting aged BBAs, which were typically considered as hygroscopic, may have limited water uptake capability.

#### 1 Introduction

Biomass burning aerosol (BBA) particles originated from wildfire burning are an important atmospheric aerosol component. The Canadian wildfires in 2023 have been unprecedented due to the scale and intensity, with a record-breaking burned area of approximately 15 million hectares. The estimated carbon emission is approximately 647 TgC, comparable to the annual fossil fuel emissions of moderate large nations (Byrne et al., 2024; Jain et al., 2024; Khaykin et al., 2025). Large amounts of

<sup>&</sup>lt;sup>1</sup>Univ. Lille, CNRS, UMR 8518 – LOA – Laboratoire d'Optique Atmosphérique, 59650 Lille, France

<sup>&</sup>lt;sup>2</sup>Prokhorov General Physics Institute (GPI) of the Russian Academy of Sciences, Moscow, Russia

<sup>&</sup>lt;sup>3</sup>Laboratoire Atmosphères, Observations Spatiales (LATMOS), UVSQ, Sorbonne Université, CNRS, IPSL, Guyancourt, France

<sup>&</sup>lt;sup>4</sup>Université de Lille, AERIS/ICARE Data and Services Center, CNRS, CNES, UMS 2877, Villeneuve d'Ascq, France **Correspondence:** Qiaoyun Hu (qiaoyun.hu@univ-lille.fr) and Philippe Goloub (philippe.goloub@univ-lille.fr)

particles and vapors are emitted into the atmosphere during wildfire combustion. Depending on the injection height, BBAs from wildfires could remain several weeks or months before being removed from the atmosphere by wet or dry deposition. Atmospheric BBA particles influence the radiation budget directly by interacting with the solar radiation budget, and indirectly by changing cloud properties and processes. Through atmospheric circulations, BBA particles can distribute over the globe and reach remote regions, influencing air quality and climate over a global scale (Schill et al., 2020). If injected into the stratosphere, BBA particles could also impact the stratospheric chemistry and deteriorate the depletion of ozone layers (Trickl et al., 2015; Ansmann et al., 2022; Ohneiser et al., 2022; Solomon et al., 2022).

Freshly emitted BBA particles rapidly enter into an aging process, involving a series of complex and competing chemical and physical processes, such as oxidation, coagulation, condensation, dilution, and evaporation. Aged BBA particles are typically composed of black carbon (BC) cores and organic carbon (OC) coatings. The BC cores represent only a few percent of the total mass, while the OC coatings dominate the particle mass (Yu et al., 2019; Czech et al., 2024). During the transport of BBA plumes, these processes continuously alter the chemical composition and the physical properties of the particles. However, our understanding of long-range transported BBA particles, for example, those involved in intercontinental transport, is still limited. Laboratory experiments face challenges in simulating the aging process over a long period. Similarly, field observations targeted on long-range transported BBA plumes are resource-intensive and limited by the uncontrollable field conditions and long distance tracking of BBA plumes. Furthermore, the aging conditions in the ambient atmosphere is more complicated, as it involves cloud processing and exposure to various precursor species during the transport.

Multi-wavelength lidar measurements have provided crucial insights into the properties and impacts of long-range transported BBAs. These lidar measurements have shown that the particle size of BBAs, after long range transport, is typically bigger than those detected near the source region (Müller et al., 2007; June et al., 2022). The increase in particle size enhances their effectiveness as cloud condensation nuclei (CCN). Lidar measurements also revealed unexpectedly high depolarization ratios in BBA layers in the upper troposphere and lower stratosphere (UTLS), in contrast to the free troposphere (FT). This could be explained by the irregular morphology of BBA particles, attributable to the aging process or the lifting mechanisms during the emission. Additionally, lidar observations provide unique information for the study of aerosol-cloud-interaction. Ice cloud formation was usually observed simultaneously with BBA plumes, suggesting that BBA particles may be effective INPs in the atmosphere (Hu et al., 2022; Mamouri et al., 2023). The organic species in BBAs, such as the Humic-like substance, polycyclic aromatic hydrocarbon (PAH) and some secondary organic aerosols (SOA) formed during the aging process, are effective fluorophores, making them good a target for laser-induced fluorescence (LIF) lidar (Lee et al., 2013; Garra et al., 2015; Zhang et al., 2019). The fluorescence capacity and the fluorescence spectrum are related to the concentration and the species of the fluorophores in the total aerosol mass. Therefore, the LIF lidar can provide a new dimension of information to aerosol characterization, as the fluorescence signal serves as a proxy to aerosol chemical composition. Currently, there are basically two types of LIF lidar systems—one utilizes single-channel broadband fluorescence channel, which can be conveniently integrated into existing lidar system, and other one detects the fluorescence spectrum with spectrometers or broadband interference filters at selected spectral range (Sugimoto et al., 2012; Reichardt, 2014; Reichardt et al., 2023, 2022; Veselovskii et al., 2023, 2020). Observations of aged BBA particles from LIF lidar systems showed particles originated from wildfires have https://doi.org/10.5194/egusphere-2025-5041 Preprint. Discussion started: 18 November 2025

© Author(s) 2025. CC BY 4.0 License.

strong fluorescence efficiency, making them distinguishable from other aerosol types and detectable even in some cloud conditions. Additionally, the fluorescence capacity and spectrum of BBAs varied case-to-case, which is probably linked with the fire source, aging condition and lifting mechanism (Reichardt et al., 2022; Veselovskii et al., 2024b). However, at current stage, the sparsity and heterogeneity of existing LIF lidar systems, as well as the gap between field observation, particularly remote sensing observations, and laboratory measurements hinder the understanding of aged BBA properties, their aging process and role in the atmosphere.

In this study, we analyzed the observations of long-range transported BBA plumes from Canadian wildfires in 2023 with two LIF lidar systems at different locations—a multi-wavelength Mie-Raman-polarization lidar equipped with a single broadband fluorescence channel in France and a 5-channel fluorescence lidar in Russia. The two lidar systems are located in the downwind of long-range transported BBA plumes from Canada. In 2023, both lidar systems accumulated a rich dataset of BBA observation due to the long wildfire season lasting from May to September. The dataset provides complementary information about BBA properties, allowing to bridge the gap caused by lidar configurations. The description of two lidar systems is presented in Section 2, and followed by the case analysis, where important features of BBA particles are demonstrated case-by-case. Additionally, we present a cross-comparison of fluorescence measurements between four lidar systems and the statistical results of BBA properties recorded in the 5-month observation in 2023, providing a comprehensive characterization of BBA particles with LIF lidar observations.

#### 2 Observation site and lidar system

Lidar observations presented in this study are obtained from two lidar systems operated at two different sites – ATOLL observatory (50.611N, 3.142E), France and GPI (General Physics Institute of the Russian Academy of Sciences, 55.235N, 37.548E) in Moscow, Russia. The distance between the two sites is about 2300 km. Influenced by the polar jet stream and pressure systems over the Atlantic Ocean, air mass transport from North America to the two lidar stations follows typically 2 pathways. The first is a more zonal (west-to-east) flow across the North Atlantic, reaching the ATOLL observatory, while the second is shifted northeastward, passing through the polar region and/or the Scandinavia to reach Moscow. The influence of polar jet stream results in cooler and drier air masses reaching Moscow, whereas the air masses arriving at ATOLL may pick up moisture as they traverse the Atlantic Ocean (Barry and Chorley, 2009; Wallace and Hobbs, 2006). Figure 1 shows the geographical locations of the two sites and the 7-day back trajectories of air mass at 5000 m reaching the two sites, dating back from 20:00 UTC, 14 May 2023, which is the onset of the BBA observation for both sites in 2023.

#### 2.1 Lidar at ATOLL, France-LILAS

The lidar system– LILAS, operated at ATOLL observatory which is a National Facility affiliated to ACTRIS, has a Nd:YAG laser source emitting at 355, 532 and 1064 nm with corresponding pulse energy of 100, 90 and 100 mJ. The laser has a repetition rate of 20 Hz. The backscattered light is collected with a Newtonian telescope of 40 cm diameter. The reception

**Figure 1.** The locations of lidar systems at ATOLL observatory(50.611N, 3.142E, 60 m a.s.l), France and Moscow, Russia (55.235N, 37.548E, 87 m a.s.l). The red and black solid lines represent the 7-day HYSPLIT back trajectories for ATOLL and Moscow, starting from 20:00 UTC, 14 May 2023. The base map is taken from ©Google Earth.

module includes detection channels for the three elastic wavelengths, each equipped with a pair of cross- and co- polarization channel, and three Raman wavelengths: 387 ( vibrational Raman of  $N_2$ ), 408 (vib-rotational Raman  $H_2O$  vapor) and 530 nm (rotational Raman of  $N_2$  and  $O_2$ ). Additionally, LILAS has a broadband fluorescence channel of 44 nm width centered at 466 nm. The lidar signals are digitized with Licel Transient recorders, allowing for a simultaneous acquisition of an analog and photon-counting signal (except for 1064 nm channel, which has only analog detection), with a range resolution of 7.5 m. This configuration allows the acquisition of vertical profiles of the  $2\alpha + 3\beta + 3\delta$  ( $\alpha$ : extinction coefficient,  $\beta$ : backscatter coefficient,  $\delta$ : particle linear depolarization ratio) dataset, the spectral fluorescence backscattering coefficient and fluorescence capacity, as well as the water vapor mixing ratio (WVMR) and the relative humidity (RH). The fluorescence backscattering coefficient is computed in a way similar to the calculation of WVMR and requires a calibration constant, as expressed by Equation (1):

$$\beta_{F,\lambda}(z) = \frac{C_R}{C_{F,\lambda}} \frac{P_{F,\lambda}(z)}{P_R(z)} \frac{T_R(z)}{T_{F,\lambda}(z)} N_R(z) \sigma_R,\tag{1}$$

where  $P_F(z)$  and  $P_R(z)$  are the lidar signals of fluorescence and Raman channels, respectively.  $\frac{C_R}{C_F}$  is the calibration constant, which is determined by the ratio of the instrumental constant of the Raman channel to the fluorescence channel.  $N_R(z)$  is the vertically distributed number concentration of Raman scatters and  $\sigma_R$  is the Raman differential scattering cross section in the backward direction.

The calibration constant accounts for the ratio of the opt-electronic efficiency of the fluorescence channel to the nitrogen Raman channel. And it can be determined by swopping the PMTs of the two channels (See Appendix A). The fluorescence capacity is the ratio of the fluorescence backscattering coefficient to the elastic backscattering coefficient at the fluorescence excitation wavelength (i.e., 355 nm for LILAS system). The fluorescence capacity of aerosol particles is mainly linked to their chemical composition, representing the capacity of aerosols in producing fluorescence signals when exposed to radiation. More detailed

description about the definition and calculation of fluorescence backscattering and capacity can be found in Veselovskii et al. (2020). To facilitate the comparison of aerosol fluorescence properties derived from different lidar systems, which may adopt interference filters of different width and spectral range, in this study, we redefine fluorescence backscattering/capacity as spectral fluorescence backscattering/capacity, denoted as  $B_{F,\lambda}/G_{F,\lambda}$ , by dividing the fluorescence backscattering/capacity by the width of the interference filter, as expressed in below:

$$B_{F,\lambda}(z) = \frac{\beta_{F,\lambda}(z)}{\Delta D_{\lambda}},\tag{2}$$

$$G_{F,\lambda}(z) = \frac{B_{F,\lambda}(z)}{\beta_{355}(z)},\tag{3}$$

where  $\beta_{F,\lambda}$  represents the spectral integrated fluorescence backscattering coefficient and  $\Delta D$  represents the full-width-half-maximum (FWHM) of the interference filters in the fluorescence channels.

#### 2.2 Five-channel fluorescence lidar at Moscow, Russia

The lidar operated at GPI in Moscow, Russia utilizes a tripled Nd: YAG laser at 355 nm, with pulse energy of 80 mJ and repetition rate of 20 Hz. To avoid the contamination to the fluorescence measurements, the laser radiation at 532 and 1064 nm is redirected by a dichroic mirror and then cleared by an optical dump. The characteristics of the telescope and the data acquisition recorder are the same with the LILAS system. The optical reception module consists of an elastic channel at 355 nm, a  $N_2$  Raman at 387 nm and five fluorescence channels respectively centered at  $438(\Delta D=29)$ ,  $472(\Delta D=32)$ ,  $513(\Delta D=29)$ ,  $560(\Delta D=40)$ , and  $614(\Delta D=54)$  nm. This configuration allows for the detection of the extinction and backscattering coefficient at 355 nm, the spectral fluorescence backscattering coefficients and fluorescence capacities at five channels. The calibration of the 438 nm fluorescence channel is performed by swapping the PMTs, as described in the Appendix. And the relative sensitivity of the rest fluorescence channels with respect to the 438 nm channel is determined using a tungsten–halogen lamp, Thorlabs QTH10/M, with a color temperature of 2800 K. A more detailed description about this lidar system and the calibration procedure is presented in the study of Veselovskii et al. (2023). The spectral fluorescence measurements provide valuable information about the chemical composition of aerosols. In this study, we use the color ratio (CR<sub>560/472</sub>) between the 560 and 472 nm channel to represent the spectral variation of aerosol fluorescence:

130 
$$CR_{560/472}(z) = \frac{B_{F,560}(z)}{B_{F,472}((z)}$$
 (4)

# 3 Case analysis

120

125

135

Four representative lidar observation cases, including two cases from ATOLL and two from GPI, were selected for detailed analysis. These observations capture the typical characteristics of BBA properties and demonstrate their variability, likely influenced by the fire origin, injection mechanism in the source region, and atmospheric processing during the long-range transport. The observations were measured in May and June, 2023, during the peak wildfire activity, which allows the plume origins to be identified with higher confidence. Moreover, the BBA layers were thick, reducing the possibility of contamination by residual

150

155

layers from earlier burning events.

#### 3.1 Case 1: 14 May 2023 at ATOLL, France

On 14 May 2023, LILAS detected a thick BBA plume beneath a dense cloud layer extending from 6000 to 12000 m height, as shown in Figure 2. According to the back trajectory analysis, the plume originated from wildfire on 5-6 May (shown in Figure B1 Appendix), in Alberta in western Canada. This observation suggests the Canadian fires had intensified, as LILAS had only detected some thin and drifting BBA layers before this date. The structure of the BBA layer is clearly illustrated in the quicklook of fluorescence backscatter coefficient in Figure 2(c), which shows the base of the BBA layer at 4000 m and the top at 6000-7000 m, in contact with the cloud base. A very thin BBA layer was observed at around 12000 m height, shown by an enhancement in the fluorescence backscatter. However, the thick mixed phase and cirrus clouds at 6000-12000 m height resulted in significant noise in the fluorescence backscattering signal.

Figure 3 exhibits the profiles of BBA properties averaged between 21:00 and 22:00 UTC and relative humidity obtained from both lidar observation and the ERA-5 reanalysis. The cloud base was stable at around 6200 m and the fluorescence backscatter coefficients extended up to 6800 m, indicating the presence of BBA particles inside the cloud base, where the humidity is high. Inside the BBA layer at 4600 to 6000m, lidar ratios are about 36 and 69 sr at 355 nm and 532 nm and particle linear depolarization ratios at 355, 532 and 1064 nm are respectively  $0.08\pm0.01$ ,  $0.05\pm0.01$  and  $0.013\pm0.002$ , showing a typical spectral dependence of BBAs (Hu et al., 2019; Haarig et al., 2018; Baars et al., 2019). The extinction and backscatter related Angström exponents are approximately  $0.7\pm0.2$  and  $2.2\pm0.2$ . The fluorescence signal is strong inside the layer, and spectral fluorescence capacity is approximately  $3.4\pm0.2 \times 10^{-6}$  nm<sup>-1</sup>, which is an important feature of BBA aerosols due to the abundance of fluorescent molecules formed during the combustion of biomass.

Both ERA-5 and lidar-derived RH profiles show a steep increase from about 30% at 4000 m to nearly 90%–100% at the cloud base. However, the vertical oscillations appearing in the RH profile from lidar measurement were not observed in the ERA-5 data, likely due to the decreasing vertical resolution with increasing height in ERA-5. As RH increases, hygroscopic aerosol particles absorb water from the surrounding air. This water uptake by particles is a rapid process—typically reaching equilibrium within seconds. Therefore, it is often considered as an equilibrium process rather than time-dependent (Carslaw, 2022). Water uptake increases particle size and sphericity, which can be detected in lidar measurements through increased elastic backscattering and a reduction in the depolarization ratio (Navas-Guzmán et al., 2019; Dawson et al., 2020; Veselovskii et al., 2024a). In this case, despite a sharp RH increase above 5000 m, the backscatter coefficients at 355 and 532 nm remained steady or even decreased near the cloud base, showing no enhancement of elastic scattering due to water uptake. Additionally, the particle linear depolarization ratio at 532 nm remained low (around 0.05), and the EAE increased only slightly with height, suggesting no obvious response to the RH. Similarly, the variations in lidar ratios and BAE with increasing RH are more attributable to the vertical variability of BBA properties other than hygroscopic growth. The fluorescence capacity, which is sensitive to hygroscopic growth, as shown by Veselovskii et al. (2024a) in an urban aerosol layer, exhibited minor changes near the cloud base. These observations suggest that BBA particles not showing significant hygroscopic growth even at RH levels of 90%–100%.



The fluorescence quenching—i.e. the suppression of aerosol fluorescence capacity due to environment factors, for instance the humidity—has been reported in several publications (Reichardt et al., 2025; Gast et al., 2025). However, we did not detect this phenomena in the present case, although it was detected in other events during the 2023 fire season (Veselovskii et al., 2024a). The absence of fluorescence quenching under high RH conditions remain an open question and requires further observational and laboratory investigations.

Previous studies have claimed that freshly emitted BBAs contain essentially non-hygroscopic compounds, however, they become increasingly hygroscopic during the aging process, due to the increase of oxidation level (Rudich et al., 2007; Jimenez et al., 2009; Massoli et al., 2010; Lambe et al., 2011). Nevertheless, the assessment of BBA hygroscopicity is difficult because of the complex and widely varied of organic compounds. Our findings in this case contribute to the relatively limited evidence showing that aged BBA particles originated from Canadian wildfires do not exhibit significant hygroscopicity. Similarly, Zheng et al. (2020) found that long-range transported smoke layers descending to the marine boundary layer have substantially lower hygroscopicity compared to the background marine aerosol. It is worth noting that the influence of aerosol fluorescence on lidar water vapor measurement cannot be completely excluded, particularly in BBA layers whose fluorescence spectrum touches the water vapor Raman wavelength at around 407.5 nm. Two strategies can be adopted to mitigate this effect: either performing a correction to subtract the aerosol fluorescence contribution from water vapor signal, or reducing the bandwidth of water vapor interference filter to minimize the fluorescence contamination (Reichardt et al., 2023; Veselovskii et al., 2024a). However, the correction approach is not feasible for LILAS measurements, as it requires at least two fluorescence channels to characterize the spectral behavior of aerosol fluorescence. Instead, LILAS utilizes a very narrow interference filter of 0.3 nm bandwidth at 407.5 nm, to suppress as much as possible the influence of fluorescence. Considering that the humidity derived from measurements during the fire season in 2023 did not exhibit anomalous values in the tropospheric BBA layers, we can reasonably assume the BBA fluorescence had quite limited impact on the measurements of LILAS, although it could increase, to some extent, the uncertainty of the absolute values. To ensure the robustness of our analysis, RH profiles from ERA-5 reanalysis and radiosonde data were used to validate the lidar-derived humidity measurements.

Figure 2. Lidar observations at ATOLL observatory in the night of 14 May 2023. (a) The backscatter coefficient (unit:  $m^{-1}sr^{-1}$ ) at 532 nm. (b) The uncalibrated fluorescence backscattering coefficient at 466 nm and (c) the volume linear depolarization ratio at 532 nm. The white pixels on the images are negative values resulted from the low signal-noise-ratio above thick clouds.

**Figure 3.** Vertical profiles of (a) backscatter coefficient and particle linear depolarization ratio, (b) spectral fluorescence backscatter coefficient and spectral fluorescence capacity, (c) RH from lidar and ERA-5, (d) Angström exponent: EAE and BAE, and (e) lidar ratios. The profiles of BBA properties and RH (red triangle line) are calculated from lidar measurements averaged between 21:00 and 22:00 UTC, 14 May 2023. The RH profiles from ERA-5 are at 21:00 and 22:00 UTC, 14 May 2023.

# 195 3.2 Case 2: 27-28 May 2023 at ATOLL, France



The BBA layers detected at ATOLL on 27 and 28 May originated from wildfires on 19 and 20 May in Alberta, British Columbia and Saskatchewan, in the western area of Canada. Giant and dense smoke plumes which can be observed from MODIS observations in Figure B1(b). In the night of 27 to 28 May, lidar quicklooks revealed stratified BBA plumes over ATOLL at 2000 to 12000 m height, as shown in Figure 4. One notable feature is that BBA layers at 12000 m showed higher depolarization ratio at 532 nm and fluorescence capacity than those in the free troposphere, as shown in Figure 4(b) and (d).

BBA optical properties derived from averaged lidar observations between 20:55 and 22:15 UTC are plotted in Figure 5. The layers with extinction and backscatter coefficients peaking at 4000 m and 12000 m are identified as BBAs, due to their specific signatures in lidar ratios, depolarization ratios and their capability of producing fluorescence when exposed to laser radiation. Between 7000 and 10000 m, an optically thin residual layer was observed in the profiles of elastic and fluorescence backscattering, although it was near the detection limit of LILAS. In the planetary boundary layer (PBL), below 2000 m, background aerosols dominated. Additionally, the RH from lidar measurement and ERA-5 dataset both show air mass was dry in these two BBA layers, with RH lower than 40%, therefore, no hygroscopic effect is observed.


Figure 4. Lidar observations at ATOLL observatory between 20:30 UTC and 03:00 UTC in the night of 27 to 28 May 2023. (a) The backscatter coefficient (unit:  $m^{-1}sr^{-1}$ ) at 532 nm. (b) The uncalibrated fluorescence backscattering coefficient at 466 nm and (c) the volume linear depolarization ratio at 532 nm.

Table 1 summarizes the optical properties of two BBA layers at different vertical levels. The BBA layer at around 12000 m exhibited mean linear depolarization ratios of 0.20. 0.14 and 0.03 at 355, 532, and 1064 nm, respectively, while the depolarization ratios in the BBA layer at around 4000 m are about 50% lower at each wavelength. Such a difference of depolarization ratio has been detected in previous lidar observations of transported BBA layers. The EAE (extinction-related Angström exponent) of approximately 0.0 is also a characteristic of aged BBA in the UTLS, whereas, typical BBA in the middle or lower troposphere tend to have slightly higher EAE. Notably, in this case we observed a notably higher fluorescence capacity in the UTLS BBA layer than the tropospheric BBA layer, adding a new evidence that UTLS BBAs differ from those in the free troposphere in microphysical and chemical properties.

**Figure 5.** Vertical profiles of (a) extinction coefficient (at 355 and 532 nm) and extinction-related Angström exponent, (b) backscatter coefficient (at 355, 532 and 1064 nm) and backscatter-related Angström exponent, (c) lidar ratios (355 and 532 nm), (d) particle linear depolarization ratios (at 355, 532 and 1064 nm), (e) the spectral fluorescence backscatter coefficient and spectral fluorescence capacity, and (f) relative humidity. The square plots and error bars represent the mean values and standard deviations in the UTLS layer at 11900 m to 12300 m. The lidar observations are averaged between 20:55 and 22:15 UTC, 27 May 2023 and the ERA-5 meteorological data is at 22:00 UTC, 27 May 2023.

**Table 1.** Optical properties of BBA particles and RH in different vertical ranges in Case 1 (14 May 2023) and 2 (27 May 2023), observed at ATOLL observatory. The means and standard deviations in three BBA layers: 4600-6000 m (Case 1), 3500-5000 m and 11900-12300 (Case 2) m are computed and summarized in the table. Be noted that the values after  $'\pm'$  represent the standard deviation in the vertical range.

| Date   | Height<br>[m] | EAE           | BAE     | LR <sub>355</sub><br>[sr] | LR <sub>532</sub><br>[sr] | PLDR <sub>355</sub> | PLDR <sub>532</sub> | PLDR <sub>1064</sub> | $G_{F,466}$ $10^{-6} [nm^{-1}]$ | RH<br>[%] |
|--------|---------------|---------------|---------|---------------------------|---------------------------|---------------------|---------------------|----------------------|---------------------------------|-----------|
| 14 May | 4600–6000     | 0.7±0.2       | 2.2±0.2 | 36±4                      | 69 ±5                     | 0.08±0.01           | 0.05±0.01           | 0.013±0.002          | 3.4±0.3                         | 30–100    |
| 27 May | 3500-5000     | 0.4±0.2       | 2.3±0.1 | 38±5                      | 81 ±12                    | 0.09±0.02           | 0.06±0.01           | 0.013±0.002          | 2.7±0.3                         | 20        |
|        | 11900-12300   | $0.0 \pm 0.3$ | 1.8±0.1 | 30±7                      | $66 \pm 12$               | $0.20 \pm 0.03$     | $0.14 \pm 0.01$     | $0.026 \pm 0.005$    | $7.0 \pm 0.6$                   | 30        |







#### 3.3 Case 3: 31 May–01 June, 2023, at GPI, Russia

In the night of 31 May to 01 June 2023, BBA plumes transported to the GPI site also originated from western Canada, similar to Case 2. The emission of the plumes dated back to 27 May, 2023, according to HYSPLIT back trajectory (See Figure B2(a)). The lidar quicklooks in Figure 6 show BBA plumes, marked by strong fluorescence signals, were distributed at 4000 m to 10000 m height. The PBL height was at around 2000 m, with a residual layer suspending above it. Two BBA plumes sequentially appeared in the height range of 9000–10500 m, and the second plume presented at 01:40 UTC showed a stronger fluorescence signal.

Aerosol properties averaged in two time intervals across the night of 31 May to 01 June are plotted in Figure 7 and summarized in Table 2. The first time interval (T1) is from 22:30 to 23:58 UTC, 31 May 2023, and the second (T2) from 01:30 to 02:30 UTC, 01 June 2023. For clarity, only the spectral fluorescence backscatter coefficient at 513 nm and the elastic backscatter coefficient, as well as the vertically averaged fluorescence capacities in the second time interval are plotted for comparison with the first time interval. The color ratio of fluorescence signals at 560 to 472 nm, CR<sub>560/472</sub> in Figure 7(a) shows higher values in the UTLS layer than in the tropospheric layer. Similar to Case 2, the lidar ratio in the UTLS layer was about 36 sr, lower than 55 sr in the tropospheric BBA layer. The relative humidity at 4000 to 7000 m was in the range of 10%-50%, according to radiosonde and ERA-5 data. In the UTLS layer, radiosonde measurements provided RH values around 50%, noticeably drier than the prediction of ERA-5, i.e, 60%-80%. According to our analysis, higher RH values of ERA-5 reanalysis than radiosonde/lidar measurements are often detected in the UTLS for both ATOLL and GPI station during the wildfire season in 2023. The discrepancy between model and radiosonde data in the UTLS has been reported in several previous studies and may be related to the lack of radio sonde measurements above the upper troposphere or/and the bias in other parameters in the meteorological field, for example, the temperature (Simmons et al., 2020; Sun et al., 2021; Krüger et al., 2022). The radiosonde sensor, for example, the RS92 sonde, has also well-known bias, which could underestimate RH in daytime measurements, however, at night, it showed much smaller errors (Bock et al., 2013). Therefore, in this study we take the radiosonde data as reference when it diverges from ERA-5 data.

The spectral fluorescence capacities are averaged within three vertical ranges in the PBL, FT and UTLS, and are plotted in Figure 7(d). In both time intervals, the spectral fluorescence capacities in the PBL decreased with wavelengths. In the 5000–6000 m, the fluorescence capacities of BBA particles increased significantly at wavelengths greater than 438 nm and peaked at 513 nm, with the values of approximately  $7.3 \times 10^{-6}$  nm<sup>-1</sup> in the first time interval and  $8.1 \times 10^{-6}$  nm<sup>-1</sup> in the second period. The UTLS layer in the first time interval showed even lower spectral fluorescence capacities than in the FT at wavelengths shorter than 560 nm. From the first to the second time interval, the BBA layer in the FT showed minor changes in the values and the spectral dependence. In contrast, the UTLS layer exhibits significantly stronger fluorescence capacities in the second time interval than in the first interval. Whereas, their normalized spectral fluorescence capacities, shown in Figure7(e), are still in good agreement, both showing a red shift in the peak toward 560 nm in UTLS. This red-shift of BBA fluorescence spectrum in UTLS was also detected by Reichardt et al. (2025) at Lindenberg, Germany in transported BBA plumes from Canadian wildfires in 2023. It confirms that this was not a feature detected during specific events, but a recurring feature of transported

**Figure 6.** Lidar observations at GPI, Moscow, Russia between 22:30 UTC and 02:30 UTC in the night of 31 May to 01 June 2023. (a) Backscatter coefficient at 355 nm, (b) spectral fluorescence backscatter coefficient at 513 nm and (c) the volume linear depolarization ratio at 355 nm.

**Figure 7.** Vertical profiles of (a) spectral fluorescence backscatter coefficients (at 438, 472, 513, 560 and 614 nm) and color ratio of 560 to 472 nm, (b) extinction and backscatter coefficients and lidar ratio at 355 nm, (c) relative humidity from radiosonde measurements (00:00 UTC, 01 June) and ERA-5 reanalysis (23:00 UTC 31 May and 02:00 UTC, 01 June). The temporal average was performed in two time intervals, the first (T1, solid line) is from 22:30 to 23:58 UTC, 31 May 2023, and the second (T2, dashed line) from 01:30 to 02:30 UTC, 01 June 2023. (d) The spectral fluorescence capacities and (e) normalized spectral fluorescence capacities averaged within three vertical ranges: 800–1200 m in the PBL (powder blue), 5000–6000 m in the FT (red) and 8500–9200 m (T1)/9600–10000 m (T2) in the UTLS (cyan). For clarity, not all profiles from T2 are shown.

BBA plumes during the Canadian wildfire season in 2023.

The BBA plumes detected in Case 2 and 3 shared similarities in geographical locations of fire sources and closeness of detection time, which offers a good opportunity for the comparison of fluorescence measurements. In Case 3, the spectral fluorescence capacity in the 472 nm fluorescence channel is in the same order of magnitude with the 466 nm fluorescence channel in Case 2 (see Table 1 and 2). While Case 2 showed a markedly higher fluorescence capacity of the BBA layer in the UTLS than in the FT, observations in Case 3 suggest that BBA particles in the UTLS do not necessarily always exhibit higher fluorescence capacity. Instead, the redshift of the fluorescence spectrum is a more recurring feature that BBA layers in UTLS differ from those in the FT.



#### 3.4 Case 4: 20 June 2023 at GPI, Moscow, Russia

Wildfires in Canada intensified significantly from the beginning of June 2023, with active fires spreading from northern British Columbia, Alberta and Saskachewan to central Alberta and the southern region of Northwest Territories. In the same period, large-scale wildfires broke out in eastern Canada, particularly in Quebec, making substantial contributions to the emission of BBA particles into the atmosphere. MODIS observations (Figure B2(b)) show the extensive coverage of BBA plumes stretching from western to eastern Canada on 13 June, making the attribution of individual plumes highly uncertain. According to HYPLIT back-trajectory analysis in Figure B2(b), the BBA plumes arriving at GPI station on 20-21 June likely originated from western (ex. the Alberta) and western Canada (ex. Quebec) after approximately 6 to 9 days of transport.

Figure 8 presents the lidar observations during the night of 20 to 21 June, 2023. The BBA layers extending from 3000 to 10000 m are more clearly identified in the quicklook of the fluorescence backscattering at 513 nm, compared to the elastic backscattering signal at 355 nm. The BBA layer above 10000 m is optically denser than the layers below, and marked with high fluorescence and moderate volume linear depolarization ratio. In Figure 8(c), some data points at around 10000 m show volume depolarization ratio close to 0.10 between 23:10 and 23:50 UTC. It is likely an indication of ice crystals formed inside or below the BBA layers, which has been quite often observed during BBA observations.

Figure 9 presents the BBA properties derived from averaged lidar observations and the RH profiles from ERA-5 analysis and from radiosonde measurement. The profiles of the spectral fluorescence backscatter coefficients in five channels, in Figure 9(a), show that aerosol layers from the PBL to the tropopause at around 12000 m presented different levels of fluorescence. The fluorescence backscatter coefficients, the extinction and backscatter coefficients at 355 nm peaked at 10500 m, where a thick BBA layer was detected. The color ratio CR<sub>560/472</sub>, showed a clear increase versus height, from below 0.6 in the PBL to 1.2 near the tropopause. The increasing trend is particularly strong in the thin BBA layer at 7000 to 9000 m.

Lidar ratios at 355 nm calculated for the two BBA layers, 7000-9000 m and 10300–10800 m, are approximately 38 sr and 32 sr, respectively, which are in good agreement with the observations in Case 2 at ATOLL station, although lower than the values observed in the night of 31 May to 01 June, 2023 (Case 3). Additionally, both Case 2 and Case 3 demonstrate lower lidar ratios (at 355 and 532 nm for ATOLL observations, at 355 nm for GPI observations) in the UTLS layers than in the free tropospheric layer. This signature could be an indicator of different morphology and/or radiative properties of BBA particles in the UTLS and in the troposphere.

Radiosonde measurements at Moskva station indicated RH values generally below 40% at above 2000 m, whereas ERA-5 reanalysis showed RH increasing above 7000 m to a peak of 90% near 10500 m. Despite uncertainties in RH estimates from both radiosonde measurements and model data, we observed no evidence of hygroscopic growth in the BBA layer. Therefore, we can conclude that BBA properties in this case are not significantly influenced by the humidity in the atmosphere.

The spectral fluorescence capacities in five channels and the normalized spectra, averaged in three vertical ranges, are plotted in Figure 9(d) and (e). Between 800 and 1200 m, where urban aerosol was the dominant aerosol type, the spectral fluorescence capacity monotonically decreased with wavelength–from  $1.50\pm0.12\times10^{-6}$  nm<sup>-1</sup> at 438 nm to  $0.43\pm0.04\times10^{-6}$  nm<sup>-1</sup> at 614 nm. In contrast, within the BBA layer between 7000 and 8000 m, the spectral fluorescence capacity increased toward longer

**Figure 8.** Lidar observations at GPI station in Moscow, Russia between 23:00 UTC and 03:00 UTC in the night of 20 to 21 June 2023. (a) Backscatter coefficient at 355 nm, (b) spectral fluorescence backscatter coefficient at 513 nm and (c) the volume linear depolarization ratio at 355 nm.



**Figure 9.** The profiles of BBA properties derived from lidar observations and RH from ERA-5 and radiosonde measurements. (a) The spectral fluorescence backscatter coefficients and color ratio between 472 and 560 nm, (b) extinction and backscatter coefficients at 355 nm, (c) RH from ERA-5 and radiosonde data, (d) the spectra of fluorescence capacity and (e) corresponding normalized fluorescence capacity at 472 nm. The lidar observations were conducted between 00:50 and 02:30 UTC on 21 June 2023 and the RH profiles are at 00:00 UTC 21 June 2023.

wavelengths, peaking at approximately  $1.7 \times 10^{-6}$  nm<sup>-1</sup> around 472 and 513 nm. In the higher BBA layer (10300 –10800 m), the spectral fluorescence capacities across the five channels increased significantly, with the spectral peak shifting further to longer wavelengths. The maximum fluorescence capacity reached  $6.0\pm0.7\times10^{-6}$  nm<sup>-1</sup> at 513 nm, closely followed by a second maximum of  $5.8\pm0.8\times10^{-6}$  nm<sup>-1</sup> at 560 nm. The shift of the fluorescence spectrum is further evidenced by the increase of color ratio,  $CR_{560/472}$  in this layer, as shown in Figure 9(a).

Table 2 presents a summary of the BBA properties observed in Case 3 and Case 4. The fluorescence channel at 472 nm of the GPI lidar is spectrally close to the 466 nm fluorescence channel of LILAS lidar at ATOLL observatory, therefore these two channels are used here for comparison. The spectral fluorescence capacities (at 472 and 466 nm) presented in the four cases are comparable in magnitude. Particularly in Case 2 and Case 4, both the tropospheric and UTLS layer show consistent fluorescence capacities measured by two different lidar systems. Higher fluorescence capacities in the UTLS layer were detected in Case 2, Case 4 and the second time interval of Case 3 at 472 and 466 nm. An exception occurred in the first time interval of Case 3, where the UTLS BBA plume did not exhibit enhanced fluorescence capacity, suggesting that this property is variable and likely influenced by multiple factors. In contrast, a redshift in the fluorescence spectrum within the UTLS was consistently

**Table 2.** Summary of BBA properties presented in Case 3 (31 May–01 June 2023) and Case 4 (20–21 June 2023). The results in the table are plotted in Figure 7 and Figure 9. Note that the values presented before and after  $'\pm'$  represent the mean and standard deviation in the height range.

| Date             | Height range [m] | LR <sub>355</sub><br>[sr] | $G_{F,438}$ $10^{-6}[nm^{-1}]$ | $G_{F,472}$ $10^{-6}[nm^{-1}]$ | $G_{F,513}$ $10^{-6} [nm^{-1}]$ | $G_{F,560}$ $10^{-6}[nm^{-1}]$ | $G_{F,614}$ $10^{-6}[nm^{-1}]$ | RH<br>[%] |
|------------------|------------------|---------------------------|--------------------------------|--------------------------------|---------------------------------|--------------------------------|--------------------------------|-----------|
| 31 May, night    | 5000-6000        | 55±13                     | 4.8±0.5                        | 6.9±0.6                        | 7.3±0.7                         | 6.4±0.8                        | 4.0±0.6                        | 20        |
|                  | 8500-9200        | 36±7                      | $3.2 \pm 1.7$                  | $5.3 \pm 0.9$                  | $6.7 \pm 1.3$                   | $7.1 \pm 1.6$                  | 5.3±1.2                        | 50-80     |
| 01 June, morning | 5000-6000        | 54±7                      | 5.1±0.7                        | 7.9±0.7                        | 8.1±0.7                         | 6.8±0.7                        | 4.3±0.4                        | 20        |
|                  | 9600-10000       | 39±5                      | $4.8 {\pm} 2.0$                | $8.0 \pm 1.3$                  | 9.7±1.4                         | $10.8 \pm 1.6$                 | $8.4{\pm}1.1$                  | 50-80     |
| 20-21 June       | 7000-8000        | 38±13                     | 1.4±1.2                        | 1.7±0.5                        | 1.7±0.5                         | 1.3±0.6                        | 0.9±0.4                        | 38        |
|                  | 10300-10800      | 33±7                      | $3.2 \pm 1.7$                  | $5.9 \pm 0.7$                  | $6.0 \pm 0.7$                   | $5.8 {\pm} 0.8$                | $4.6 \pm 0.5$                  | 35        |

observed in Cases 3 and 4.





#### 4 Statistics and discussion

# 4.1 Extinction- and backscatter-related Angström Exponent

The extinction coefficients and Angstöm exponents of 34 BBA layers detected from May to September 2023 at ATOLL observatory are presented in Figure 10. These layers were identified as BBA based on their signatures in lidar ratios, particle linear depolarization ratios and fluorescence capacity. Among the 34 BBA layers, 26 layers were classified as free tropospheric layers (layer top < 8000 m), and the other 8 layers were classified as UTLS layers (layer base > 8000 m). The extinction coefficients of most BBA layers observed during this period were below 200 Mm $^{-1}$ . The extinction-related Angström exponent in free tropospheric layers averaged  $0.8\pm0.3$ , generally bigger than those observed in the UTLS layers, where the values averaged  $0.1\pm0.2$ . The backscatter-related Angström exponents in the free tropospheric BBA layers averaged  $2.0\pm0.2$ , slightly bigger than those averaged in the UTLS layers, which is  $1.5\pm0.2$ . Such difference of Angström exponent between tropospheric and UTLS BBAs has been reported in previous lidar observations (Haarig et al., 2018; Hu, 2018; Ohneiser et al., 2020; Mamouri et al., 2023).

Light scattering models indicate the EAE of aerosols is strongly correlated with aerosol particle size. Both field campaign and laboratory measurements showed that BBA particles tend to grow during the aging process due to the condensation of gas-phase organic compounds and particle coagulation. Particle size can also vary from fire to fire, influenced by the burning conditions, fuel types and lifting mechanisms, which can all affect the morphology and composition of BBA particles injected into the atmosphere (Selimovic et al., 2019; Hodshire et al., 2021; Perring et al., 2017; Katich et al., 2023; Kleinman et al., 2020). For example, aircraft measurements showed BBA particles injected into the UTLS by pyro-cumulonimbus (PyroCb) convections were with thicker coatings, thus resulting in bigger size compared with tropospheric BBA particles. One likely explanation is that the strong updrafts in PyroCb convections lift large amounts of aerosols and vapors, promoting the coagulation of aerosol particles and condensation of the organic vapors, which accelerates the growth of BBA particles. The aging environment in the ambient atmosphere may also play a role. For instance, particles with diameters larger than 50-100 nm are more efficient CCN particles and therefore are more likely to be washed out by wet removal, a process that is more frequent in the free troposphere (Petters et al., 2009).

In contrast to the EAE, the BAE of aerosol particles is influenced not only by the particle shape, but also by the refractive indices. A detailed investigation on the effect of these factors is beyond the scope of this study. So far, there has been few observations addressing differences in BAE between tropospheric and UTLS BBAs, underscoring the need for additional measurements to better understand this feature.

# 4.2 Lidar and depolarization ratios

The lidar ratios and particle linear depolarization ratios observed from May to September 2023 at ATOLL observatory are displayed in Figure 11. The spectral dependence of lidar ratios, which inversely correlates with wavelength, aligns with previously observations of BBAs from Canada, the US and Australia. The average lidar ratios in tropospheric BBA layers during

**Figure 10.** (a) Extinction coefficients at 355 and 532 nm, (b) Extinction-related Angström exponent and (c) backscatter-related Angström exponent measured by LILAS at ATOLL observatory from May to September 2023. Note that the BBA layer with extinction coefficient higher than 20 Mm<sup>-1</sup> are selected and classified into 2 groups: FT (free troposphere) and UTLS (upper troposphere and lower stratosphere), according to the height of the layer base.

this period are respectively 44±9 sr and 72±11 sr at 355 and 532 nm, while in the UTLS layers they were 36±4 sr and 62±4 sr. The slightly higher lidar ratios in tropospheric layers compared to the UTLS layer are consistent with lidar observation in Moscow, as shown in Case 3 and 4.

The particle linear depolarization ratios in tropospheric BBA layers averaged  $0.12\pm0.08$ ,  $0.07\pm0.03$  and  $0.02\pm0.01$  and 1064

**Figure 11.** (a) Lidar ratios at 355 and 532 nm, (b) particle linear depolarization ratios (PLDRs) at 355, 532 and 1064 nm, in the BBA layers measured by LILAS at ATOLL observatory from May to September 2023. The data points are divided into two categories by the height of the selected layers – free tropospheric (FT) layer and UTLS layer. The scatters and error bars represent the means and standard deviations in the selected BBA layers.

nm. In contrast, the corresponding values in the UTLS layers are higher at 355, 532 nm, averaging  $0.23\pm0.08$  and  $0.14\pm0.05$ , respectively, while remaining almost unchanged at 1064 nm with an average of  $0.02\pm0.01$ . Pronounced depolarization ratios of BBA layers in UTLS have been observed by several lidar systems during the remarkable wildfires in global scale – Canadian




wildfire in 2017, Australian wildfire in 2019/2020, Californian wildfire in 2020 (Haarig et al., 2018; Hu et al., 2019; Ohneiser et al., 2020; Hu et al., 2022; Mamouri et al., 2023). High depolarization ratios are usually linked with particle morphology, which can be complex for aged BBAs, as their organic coatings may appear semi-solid and glassy in cold and dry conditions in UTLS. Therefore, UTLS BBA particles could be highly irregular and present high depolarization.

The lifting process of wildfire plumes may also influence the morphology of BBA particles. Exceptionally high depolarization ratios were observed during two periods: on 12 July and from 23 to 30 September2023, with values at 355 nm approaching or exceeding 0.30. According to Khaykin et al. (2025), these plumes originated from wildfires associated with pyroCb activity, detected in Siberia on 30 June and in Canada on 15 and 22 September. PyroCb-convection can rapidly inject thick clouds of BBA particles, organic vapors, and ice crystals into the UTLS (Peterson et al., 2017). BBA particles processed by pyroCb events are found to exhibit distinctive characteristics, including larger sizes and thicker organic coatings, compared to those not affected by such convective processes (Rosenfeld et al., 2007; Katich et al., 2023). In particular, LILAS detected tropospheric BBA layers with depolarization ratios comparable to those in the UTLS on 29 and 30 September. Lidar observations revealed that these layers gradually descended from the free troposphere to the planetary boundary layer over ATOLL within 2–3 days. HYSPLIT back trajectories suggest that this descent was likely associated with a dry intrusion a strong downward motion driven by cyclonic activity that can transport air masses from the UTLS into the lower troposphere and boundary layer (Danielsen, 1968). Consequently, the observed tropospheric BBA layers with high depolarization ratios were likely of UTLS origin.

#### 4.3 Fluorescence capacity and spectrum

Transported BBA plumes originated from Alberta wildfires in late May 2023 have been reported by other lidar stations in Europe, providing an opportunity for the cross-comparison of fluorescence measurements. Figure 12 presents the spectral fluorescence capacities analyzed in Case 2 and 3 in this study, as well as measurements reported by two German lidar systems—MARTHA at TROPOS in Leipzig and RAMSES at DWD in Lindenberg. MARTHA and LILAS utilize the same interference filter at 466 nm for fluorescence measurements, while the RAMSES lidar employs spectrometers that allow for the detection of the fluorescence spectrum with a spectral resolution of about 12 nm. Consequently, the measurements of RAMSES can reveal more features of the fluorescence spectra and serve as a reference for broadband fluorescence measurements (Reichardt et al., 2023). Another important aspect of this comparison is to assess whether the fluorescence measurements, made by different lidar systems, each calibrated individually by different lidar groups, are consistent. Although the proximity in observational time does not guarantee that the plumes originated from the same wildfire, it can still eliminate the possibility of plumes coming from other regions, since the Alberta wildfires were the dominant fire sources during the second half of May.

Figure 12 demonstrates the fluorescence capacities measured by LILAS and MARTHA are in good agreement in the tropospheric BBA layer. However, in the UTLS layer MARTHA derived much lower values than LILAS. This decrease in fluorescence capacity is likely due to cloud processing, as MARTHA detected ice cloud formation at the base of the UTLS BBA layer. In the spectral range around 466 nm, the fluorescence capacities by the four lidar systems are consistent in magnitude,

390

395

**Figure 12.** Comparison of fluorescence capacities measured by four lidar systems in transported BBA plumes originated from Alberta wildfires in late May, 2023. The four lidar systems are – LILAS at ATOLL (Lille, France), MARTHA at TROPOS (Leipzig, Germany), RAMSES at DWD (Linden-berg, Germany), and the GPI lidar (Moscow, Russia). Measurements from LILAS and GPI presented respectively in Case 2 (27 May 2023, diamonds) and Case 3 (31 May – 01 June, 2023, circles with lines, solid line–time interval T1, dashed line–time interval T2) are compared with measurements from MARTHA on 29 May, 2023 (triangles) and RAMSES on 26 May 2023 (solid lines). The measurements from MARTHA and RAMSES were published by Gast et al. (2025) and Reichardt et al. (2025).

although variability should not be overlooked. At wavelengths greater than 450 nm, the spectra of RAMSES also demonstrate a gradual increase of spectral fluorescence capacities versus BBA layer height, except in the layer at 4600 m. Although spectra of the fluorescence capacities measured by GPI lidar and RAMSES show some extent of variability from layer to layer, the values are generally comparable and show consistent features, particularly in terms of the shape of the spectra and the central wavelengths in the troposphere and UTLS, which are both consistent. For instance, the two BBA layers detected by RAMSES at 5700 and 10500 m over DWD, have almost the same central wavelengths and color ratios  $CR_{560/472}$ , compared with the layers at 5500 m and 9000 m over GPI station (Case 3). A quantitative comparison of BBA fluorescence measurements is presented in Table C1 in the Appendix.

Figure 13 (a) and Figure 13(b) show the time series of spectral fluorescence capacities in BBA layers detected by lidar systems at ATOLL observatory and at GPI, respectively, in the period from May to September 2023. The average of the spectral fluorescence capacity in the tropospheric BBA layers was  $3.3\pm1.3\times10^{-6}$ nm<sup>-1</sup> at ATOLL and  $3.9\pm2.3\times10^{-6}$ nm<sup>-1</sup> GPI, while in the UTLS layers, the values were  $4.8\pm1.8\times10^{-6}$ nm<sup>-1</sup> at ATOLL and  $5.4\pm2.6\times10^{-6}$ nm<sup>-1</sup> at GPI. Despite differences in spectral coverage and calibration methods between the two fluorescence channels, the spectral fluorescence capacities at both

**Figure 13.** The time series of spectral fluorescence capacity at 460 and 472 nm, and the color ratios  $CR_{560/472}$  measured by lidar systems at ATOLL observatory (Lille, France) and GPI (Moscow, Russia) during the period from May to September 2023. (a) The spectral fluorescence capacities at 466 nm from ATOLL. (b) The spectral fluorescence capacities at 466 nm and the color ratio of fluorescence capacity/backscatter between the 560 nm and 472 nm channels, measured by the multi-channel fluorescence lidar at GPI.

sites are generally comparable in both the troposphere and the UTLS. This consistency validates the comparability of fluorescence measurements between the two different lidar stations and confirms that the properties of BBA particles arriving at these two lidar stations do not exhibit significant geographical variations. The values of spectral fluorescence capacity detected by lidar at GPI at 472 nm are generally greater than those at ATOLL observatory at 460 nm, it is probably because the 472 nm is closer to the peak of BBA fluorescence spectrum, as have been shown in Figure 12.

In Figure 13, we can see that enhanced spectral fluorescence capacities of BBA layers in the UTLS were observed at both stations. Although the spectral fluorescence capacity exhibits a substantial variability of approximately 40-60% in both the UTLS and the free troposphere, the highest values were consistently observed in the UTLS, with no comparable peaks detected in the free troposphere. The values of spectral fluorescence capacity of BBAs may depend on multiple factors, such as the geographic location of wildfires, the vegetation, the burning condition, the aging process and so on. However, it is difficult to assess their influences on the fluorescent properties of BBAs, due to the limited information derived from remote sensing observations and uncertainties in the back trajectories of the plumes. The series of  $CR_{560/472}$ , which is the color ratio of fluorescence capacity or backscatter coefficient between 560 nm and 472 nm channels, is generally higher in the UTLS, averaging  $1.3\pm0.2$ , compared to  $0.9\pm0.2$  in the troposphere. Higher color ratio in the UTLS than in the troposphere is a rather recurring feature of BBAs and has been detected consistently by both GPI and RAMSES during the fire season in 2023.

Figure 14 summarizes the spectra of fluorescence capacity, as well as their normalized forms, in aerosol layers in three ver-

**Figure 14.** (a) The spectrum of aerosol fluorescence capacity in three vertical ranges: PBL(blue line,below 3000 m), FT (cyan line between 3000 and 8000 m) and UTLS (red line, between 8000 and 12000 m), measured by the five-channel fluorescence lidar at GPI, Moscow from May to September 2023. This figure is adapted from Figure 8 in Veselovskii et al. (2024b).

tical ranges-below 3000m, from 3000 to 8000 m, and 8000 to 12000 m, measured by the lidar system at GPI from May to September in 2023. Urban aerosols, which dominated below 3000 m, showed fluorescence capacities generally lower than  $2 \times 10^{-6} \text{nm}^{-1}$ , with a spectrum monotonically decreasing versus wavelengths. Long-range transported BBAs were the major aerosol type at above 3000 m, while their fluorescence spectra show distinctive characteristics. In the free troposphere at 3000 to 8000 m, the fluorescence capacities varied between  $1.5 \times 10^{-6} \text{nm}^{-1}$  and  $5 \times 10^{-6} \text{nm}^{-1}$ . In the UTLS at above 8000 m, the mean spectral fluorescence capacities are generally higher than their counterparts in the free troposphere. Additionally, the normalized fluorescence spectra in the UTLS showed a clearly shift of the spectrum peak toward longer wavelengths.

#### 4.4 Vertical variation of BBA properties


Although the 2023 Canadian wildfire season was exceptional in terms of burnt area, fire emissions and the generated detected PyroCb count, the emitted BBA plumes were mostly limited to the upper troposphere and the lowermost stratosphere. The upper most of BBA layer heights in this study is at 12-14 km, in agreement with SAGE III (Stratospheric Aerosol and Gas Experiment) observation presented in (Khaykin et al., 2025). In this analysis, we collect three extensive parameters of BBAs – the  $EAE_{355-532}$  and particle depolarization ratio at 532 nm from the observations at ATOLL and the color ratio  $CR_{560/472}$  from the observations at GPI station, to investigate the vertical variation of BBA properties. Figure 15 presents the variations of the three parameters with respect to the altitude of the BBA layer base, along with the corresponding linear regressions.






The EAE decreases gradually with the increasing layer altitude, yielding  $r^2 \approx 0.61$ , while the depolarization ratio exhibits an increasing trend with  $r^2 \approx 0.68$ . The color ratio  $CR_{560/472}$  also shows a positive trend with altitude, with  $r^2 \approx 0.69$ . These values of  $r^2$  suggest moderate correlations; however, the data sets do not provide sufficient evidence to establish a genuine altitude dependence.

Most data points collected at ATOLL were from the free troposphere, as the calculation of EAE and depolarization ratio requires relatively higher optical thickness and signal-to-noise ratio in BBA layers, which are challenging conditions for UTLS layers. Consequently, the number of measurements in UTLS layers is lower than in the free troposphere, which limits the analysis of the vertical dependence. For the depolarization ratio in the troposphere, most data points are clustered near 0.05, with little variation across the altitude range and no clear trend versus altitude. The apparent positive correlation with altitude emerges only when the UTLS data points were included, as they are at higher altitude and exhibit distinctly higher depolarization ratios. The pattern may be an indication that the statistical correlation is driven primarily by the separation between the two regimes-tropospheric and UTLS, rather than by a continuous, altitude-driven relationship. A similar two-cluster distribution is also evident in the color ratio plot. These features are more consistent with discrete differences in BBA properties than with a vertical dependence. These observations indicate a possibility that altitude itself may not be the ultimate controlling factor of the BBA properties, but rather an intermediate variable linked to other factors, which could be the temperature and/or the injection height of the BBA layers in their source region, and so on. Reichardt et al. (2025) investigated the correlation between BBA properties and factors such as the origin, transport time, in-layer temperature and humidity, using RAMSES lidar data. They reported that the fluorescence spectra of the BBAs from different wildfire sources at different geographical locations showed, to some extent, correlation with the vegetation type (or climate zone) and in-layer temperature, while weak correlation with the transport time. Relative humidity also appeared to influence the fluorescence spectra, although its effect likely depends on the origin of BBA plumes. At present, further investigation of the dependence of BBA properties is constrained by the limited variability in the available BBA measurements and the lack of collocated and high-quality humidity and temperature data in the BBA layers.

#### 4.5 Relative humidity in BBA layers

RHs within approximately 50 BBA layers observed at each lidar station were determined using a combination of lidar and radiosonde measurements. The RH data at GPI site were obtained from radiosonde measurements at Moskva site, while at ATOLL site, priority was given to RH derived from the lidar water vapor channel. When lidar water vapor measurements were unavailable (~10 BBA layers), radiosonde data from Beauvechain (Belgium) station, 100 km from ATOLL, were used. Figure 16(a) shows the frequency distribution of detected BBA layers as a function of their mean RH. All the analyzed layers exhibited RH values below 80%, with the majority (90% at Moscow and 85% at ATOLL) showing mean RH below 50%. At both sites, observations showed that RHs generally decreased at the altitudes where BBA layers occurred, even in the free troposphere where water vapor is more abundant than in the UTLS. The peak occurrence of RH within the detected BBA layers was in the range of 10-20%.

Figure 16(b) presents the distribution of the spectral fluorescence capacity as a function of the mean RH within the BBA layers.



**Figure 15.** (a) The variation of BBA optical properties versus the height of the layer base observed by lidar systems at ATOLL and GPI from May to September in 2023. (a) The extinction-related Angström exponent at 355 and 532 nm, and (b) the particle depolarization ratio at 532 nm detected by LILAS at ATOLL observatory. (c) The color ratio of the spectral fluorescence backscatter coefficients at 560 to 472 nm, detected by the lidar system at GPI,Moscow, Russian. The error bars in the plot represent the standard deviation within the selected BBA layers. The red dot-dashed lines represent the linear regression line of the data points.

The fluorescence capacities at 472 nm (GPI lidar) and at 466 nm (ATOLL lidar) are comparable and distribute mainly in the range of  $2.0 \times 10^{-6}$  nm<sup>-1</sup> to  $4.0 \times 10^{-6}$  nm<sup>-1</sup>. Layers with pronounced fluorescence capacity, greater than  $6.0 \times 10^{-6}$  nm<sup>-1</sup>, are mainly UTLS layers, with RH approximately at 20% or lower. According to the vertically averaged data across 5 months, plotted in Figure 16(b), the spectral fluorescence capacities do not show noticeable correlation with RH when air mass is relatively dry, i.e. RH $\leq$ 60%. For RH $\geq$ 60%, the fluorescence capacity seemingly decreases with RH, however, the number of data points is not sufficient to draw a firm conclusion.

Veselovskii et al. (2025) and Reichardt et al. (2025) investigated the relationship between aerosol (not limited to BBA) fluorescence properties and RH using vertically resolved profiles from selected cases, rather than multi-month layer-averaged means. Reichardt et al. (2025) reported no clear correlation between RH and the central wavelength of the fluorescence spectra for the BBA layers likely originating from Western Canada on 26-27 May. While for BBA plumes from Eastern Canada, the results showed a weak to moderate correlation. These findings are in line with our observation in Case 1 where the fluorescence capacity stayed almost unchanged in the BBA layer with high humidity below the cloud base. While this cannot be considered direct, mutually corroborating evidence, it provides indirect support for limited hygroscopicity of aged BBAs. Most BBA layers analyzed in this study are dry and do not exhibit significant RH gradient, affecting the robustness of the correlation analysis. However, this generally low humidity in BBA layers suggests that aged BBAs have limited ability of absorbing water from the environment, implying low hygroscopicity. Fluorescence measurements have potential as a proxy for assessing aerosol hygroscopicity, but current studies are sparse and scattered, highlighting the need for more systematic investigation.

In previous publications, the atomic oxygen-to-carbon ratio (O:C) of the organic compounds are often used to assess the hy-

groscopicity organic particles, based on semi-empirical relationships between the hygroscopicity and the oxidation level of the organic aggregates (Jimenez et al., 2009; Massoli et al., 2010; Lambe et al., 2011). These researches, based on laboratory and field measurements, overall derived a generally positive correlation between the hygroscopicity and O:C ratio of organic aerosols. As a result, BBA are typically expected to become increasingly hygroscopic after being emitted into the atmosphere. However, the measured hygroscopicity of aerosols originated from biomass burning is still very variable, with  $\kappa$ —the hygroscopic parameter, varying from below 0.1 (weakly hygroscopic) to 0.4 (moderately hygroscopic) and showing dependence on multiple factors, such as fuel type and aging time (Petters et al., 2009; Carrico et al., 2010; Engelhart et al., 2012; Zheng et al., 2020; Cao et al., 2021; Pöhlker et al., 2023). Recent research reported other factors, such as carbon chain length, organic functionality and water solubility have significant impact on the hygroscopicity of organic aerosols, which explains the widely varying hygroscopicity of organic aerosols. It also points out the oxidation level of organic aerosols is not sufficient for parameterizing their hygroscopicity and the determination of the hygroscopicity of BBAs is a challenging task due to their complex organic composition and aging conditions in the atmosphere (Wang et al., 2019; Kuang et al., 2020; Han et al., 2022).

**Figure 16.** (a) The count of BBA layers and (b) their spectral fluorescence capacity as a function of mean RH within the layers observed by lidars at Moscow (blue color) at ATOLL site (gray color) from May to September 2023. The open circles and solid circles represent BBA layers in the free troposphere and UTLS, respectively. Values of spectral fluorescence capacity measured by lidar at Moscow and ATOLL are respectively at 472 and 466 nm.

#### 5 Conclusions

In this study, we analyzed the long-range transported BBA plumes from the exceptional 2023 Canadian wildfire season, using
lidar observations from ATOLL observatory in France and GPI in Russia between May and September. The complementary
detecting capabilities of the two systems, one equipped with multiple elastic and Raman channels and the other with multiple




fluorescence channels, enabled a detailed characterization of BBA properties. Four representative cases and statistical results from five-month observation were presented.

Measurements from ATOLL showed that UTLS BBAs had higher depolarization ratios (355 and 523 nm), lower Ångström exponents, and slightly lower lidar ratios than free-tropospheric BBAs, consistent with earlier studies. GPI lidar detected a redshift in the fluorescence peak from 513 nm (troposphere) to 560 nm (UTLS), confirmed by high-resolution spectra from the RAMSES lidar.

Fluorescence capacities at 466 nm (ATOLL) and 472 nm (GPI) were consistent in both tropospheric and UTLS BBA layers. We further compared the fluorescence capacities of BBA plumes measured by four lidars at different locations, including the MARTHA at Leipzig and RAMSES at Lindenberg. The results showed a general agreement in magnitude, as well as consistent spectral shape between multi-channel broadband and spectrometer-based fluorescence measurements. These results highlight the potential for coordinated multi-lidar fluorescence measurements, though further systematic comparisons are needed to fully reconcile the two detection approaches.

In-layer humidity analysis for around 100 BBA layers revealed that most BBA layers were relatively dry (RH 

Acknowledgements. This work is supported by OBS4CLIM (project number: ANR-21-ESRE-0013), CaPPA (project number: ANR-11-LABX-0005-01), ECRIN/FEDER and the ANR PyroStrat project (project number: 21-CE01-335 0007-01, https://pyrostrat.projet.latmos.ipsl.fr). In addition, the ESA/QA4EO program is greatly acknowledged for supporting the observation activity at LOA and the Russian Science Foundation is acknowledged for supporting the work at GPI through project 21-17-00114. The research activities performed at ATOLL observatory also benefit from the research infrastructure ACTRIS-FR, as well as from the Center of Aerosol Remote Sensing (CARS) of the ACTRIS Research Infrastructure. At last, we thank Dr. Jens Reichardt (Richard-Aßmann-Observatorium, Deutscher Wetterdienst, Lindenberg, Germany) and Benedikt Gast (TROPOS, Leipzig, Germany) for providing their lidar measurement for comparison.





### Appendix A: Calibration of single-wavelength fluorescence channel

The calibration of a single-wavelength fluorescence channel integrated to a Raman lidar system is essentially to determine the ratio of optical and electronic efficiency between the fluorescence channel and the Raman channel. The spectrally integrated backscatter coefficient in the fluorescence channel can be expressed by Equation 1:

$$\beta_{F,\lambda}(z) = \frac{C_R}{C_{F,\lambda}} \frac{P_{F,\lambda}(z)}{P_R(z)} \frac{T_R(z)}{T_{F,\lambda}(z)} N_R(z) \sigma_R,$$

where  $P_F(z)$  and  $P_R(z)$  are the lidar signals of fluorescence (F) and Raman channel (R), respectively.  $\frac{C_R}{C_F}$  is the ratio of the instrumental constant between the Raman channel and the fluorescence channel, depending on the optical and electronic efficiency of the two channels. This ratio is to be determined by the calibration procedure. Figure A1 shows the optical layout of the fluorescence channel and  $N_2$  Raman channel, where the transmission of the interferences filter (IF) and the efficiencies of photomultipliers are denoted as  $\alpha_{F/R}$  and  $g_{F/R}$ , respectively. The transmission of dichroic mirror (DM) splitting the two channels is denoted as  $t_{F/R}$ .

# Step 1: Keep the PMTs at their original position.

 $S_F$  and  $S_R$ , representing the lidar signals received by the fluorescence channel at 466 nm and the Raman channel at 387 nm can be written as:

$$S_F = S_0 * t_F * \alpha_F * g_F$$

$$S_R = S_0 * (1 - t_R) * \alpha_R * g_R$$
(A1)

where  $S_0$  represents the incoming light intensity at the DM.

#### Step 2: Exchange the two PMTs, without changing the IF.

Now the signals can be expressed as:

$$S'_{F} = S_{0} * t_{F} * \alpha_{F} * g_{R}$$

$$S'_{R} = S_{0} * (1 - t_{R}) * \alpha_{R} * g_{F}.$$
(A2)

The ratio of signals in Step 1 and 2 in the Raman channel at 387 nm is determined by the ratio of the gains of the two PMTs—

$$\frac{S_R}{S_R'} = \frac{g_R}{g_F} \tag{A3}$$

Note that, in Equation A1-A3, we assume the transmission in the optics before the DM is invariant between the spectral range of the Raman and fluorescence channel. Therefore, the ratio of instrument constant between the fluorescence channel and the Raman channel,  $\frac{C_R}{C_F}$ , is determined by the transmission of the DM and IFs, and by the grain of the PMTs. Since the two channels are well separated in spectrum, the DM is able to split them with high clearance, i.e.  $t_R$  =0 and  $t_F$  = 1. Therefore, the following relationship can be derived:

$$\frac{C_R}{C_F} = \frac{\alpha_R}{\alpha_F} \frac{g_R}{g_F}.$$
 (A4)


The ratio  $\frac{\alpha_R}{\alpha_F}$  can be calculated from the transmission curves of the IFs provided by the manufacturer of optics. In the system of LILAS, the mean transmission is about 65% in the Raman channel and 90% in the fluorescence channel, therefore,  $\frac{\alpha_R}{\alpha_F}$  equals approximately to 0.72. The ratio  $\frac{g_R}{g_F}$  is derived in Equation A3, following the calibration procedures in Step 1 and 2. It is also important to mention that the calibration of  $\frac{g_R}{g_F}$  specific to analog signal and photon-counting signal are performed simultaneously. The choice of  $\frac{g_R}{g_F}$ , analog  $(\frac{g_R}{g_F}|_{AN})$  or photon-counting  $(\frac{g_R}{g_F}|_{PC})$ , should correspond to the choice of channels used for the calculation of  $\beta_F$ . Additionally, if the calibration is performed correctly and the gluing coefficients, converting analog signal to photon-counting signal, are accurate,  $\frac{g_R}{g_F}|_{AN}$  and  $\frac{g_R}{g_F}|_{PC}$  should be exchangeable. For example, according to the calibration performed on LILAS system, 20 December 2023:

$$\frac{g_R}{g_E}$$
  $|_{AN}$  = 2.5 and  $\frac{g_R}{g_E}$   $|_{PC}$  = 1.5.

The gluing coefficients for 387 and 466 nm channels are approximately 65 and 105, respectively. The values of  $\frac{g_R}{g_F}|_{PC}$  can be approximated using the gluing coefficients—

$$\frac{g_R}{g_F} |_{AN} * \frac{r_R}{r_F} = 2.5 * \frac{65}{105} = 1.547 \approx 1.5.$$

Figure A1. The optical layout of the fluorescence channel at 466 nm and the  $N_2$  Raman channel at 387 nm in LILAS system at ATOLL observatory, Lille, France. The two channels are split by a dichroic mirror and neither of them is attenuated by neutral density filter. The calibration of the fluorescence channel requires to swopping the two PMTs without changing the interference filters with transmission denoted as  $\alpha$ .

# Appendix B: Back trajectory of air mass observed at ATOLL and GPI

Figure B1 and B2 plot the back trajectories of air mass arriving at ATOLL observatory and GPI stations, respectively.

NOAA HYSPLIT MODEL

**Figure B1.** The back trajectories of air mass for observations in Case 1 and 2 at ATOLL observatory, Lille, France. (a) 168-hour backward trajectory for air mass at 4000, 8000, and 12000 m height at 21:00 UTC, 27 May 2023. The base map is the true color image of MODIS on 20 May 2025. (b) 168-hour backward trajectory for air mass at 5000 and 6000 height at 22:00 UTC, 14 May 2023, overlaid on MODIS True Color surface image on 08 May 2023. The area circled by dotted orange lines were covered by intense BBA plumes (marked by yellowish or gray color), when the airmass pass through.

## **Appendix C: Comparison of fluorescence measurements**

# NOAA HYSPLIT MODEL Backward trajectories ending at 2300 UTC 31 May 23 GDAS Meteorological Data

# NOAA HYSPLIT MODEL Backward trajectories ending at 2300 UTC 20 Jun 23 GFSQ Meteorological Data

**Figure B2.** The back trajectories of air mass for observations in Case 3 and 4 measured at GPI, Moscow, Russia. (a) 120-hour backward trajectory for air mass at 6000 and 9000 m height at 23:00 UTC, 31 May 2023. The base map is the true color image of MODIS on 28 May 2025. (b) 192-hour backward trajectory for air mass at 7500 and 10500 m height at 23:00 UTC, 20 June 2023. The transport pathways are overlaid on MODIS True color surface image on 14 June 2023. The area circled by dotted orange lines represent the area covered by intense BBA plumes, marked by yellowish or gray color.

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

**Table C1.** Comparison of spectral fluorescence capacities in BBA layers from the 2023 Alberta wildfires (Canada), measured by four fluorescence lidars at different stations. The instruments include LILAS at ATOLL (Lille, France), MARTHA at TROPOS (Leipzig, Germany), RAMSES at DWD (Lindenberg, Germany), and the GPI lidar (Moscow, Russia). LILAS, MARTHA, and the GPI lidar measure fluorescence signals using broadband interference filters. LILAS and MARTHA each operate with a single fluorescence channel (44 nm bandwidth centered at 466 nm), while the GPI lidar has five discrete fluorescence channels; for this comparison, measurements at 472 nm and 560 nm were selected. RAMSES captures the full fluorescence spectrum using a spectrometer, with a spectral resolution about 12 nm (Reichardt et al., 2023). To enable comparison with broadband measurements, RAMSES fluorescence capacity was averaged over two bands centered at 495 nm and 585 nm (80 nm bandwidth), corresponding to the "cyan" and "green" channels as defined by Reichardt et al. (2025).

Abbreviations: LCH – layer central height; SPC – spectral fluorescence capacity; PWL – peak wavelength.

The color ratio is calculated as SPC\_2 to SPC\_1.

| Lidar&Location<br>&Refence | Observation  Date | LCH<br>[m] | SFC_1 [10 <sup>-6</sup> nm <sup>-1</sup> ] | SFC_2 [10 <sup>-6</sup> nm <sup>-1</sup> ] | PWL [nm] | Color ratio |
|----------------------------|-------------------|------------|--------------------------------------------|--------------------------------------------|----------|-------------|
|                            | 1                 | [111]      | [10 ]                                      | [10 1111 ]                                 | [11111]  |             |
| LILAS                      |                   |            | 466 nm                                     | _                                          |          |             |
| Lille, France              | 27-28 May         | 4000       | 2.9                                        | _                                          | _        | _           |
| This study                 |                   | 12000      | 4.4                                        | -                                          | -        |             |
| MARTHA                     |                   |            | 466 nm                                     | _                                          |          |             |
| Leipzig, Germany           | 29-30 May         | 3000       | 2.5                                        | _                                          | -        | _           |
| Gast et al. (2025)         |                   | 5000       | 4.9                                        | _                                          | _        | _           |
|                            |                   | 12000      | 6.7                                        | -                                          | _        | _           |
| GPI lidar                  |                   |            | 472 nm                                     | 560 nm                                     |          |             |
| Moscow, Russia             | 31 May, night     | 5500       | 6.9                                        | 6.4                                        | 513      | 0.93        |
| This study                 |                   | 9000       | 5.3                                        | 7.1                                        | 560      | 1.34        |
|                            | 01 June, morning  | 5500       | 7.9                                        | 6.8                                        | 513      | 0.86        |
|                            |                   | 9000       | 8.0                                        | 10.8                                       | 560      | 1.35        |
| RAMSES                     |                   |            | 472 nm                                     | 560 nm                                     |          |             |
| Lindenberg, Germany        | 26-27 May         | 3600       | 4.3                                        | 3.1                                        | 499      | 0.72        |
| Reichardt et al. (2025)    |                   | 4600       | 5.6                                        | 6.8                                        | 532      | 1.21        |
|                            |                   | 5700       | 4.8                                        | 4.7                                        | 514      | 0.97        |
|                            |                   | 10500      | 6.0                                        | 8.7                                        | 541      | 1.44        |

smoke event: decay phase and aerosol properties observed with the EARLINET, Atmospheric Chemistry and Physics, 19, 15 183–15 198, https://doi.org/10.5194/acp-19-15183-2019, 2019.

Barry, R. G. and Chorley, R. J.: Atmosphere, weather and climate, Routledge, 2009.

- Bock, O., Bosser, P., Bourcy, T., David, L., Goutail, F., Hoareau, C., Keckhut, P., Legain, D., Pazmino, A., Pelon, J., Pipis, K., Poujol, G., Sarkissian, A., Thom, C., Tournois, G., and Tzanos, D.: Accuracy assessment of water vapour measurements from in situand remote sensing techniques during the DEMEVAP 2011 campaign at OHP, Atmospheric Measurement Techniques, 6, 2777–2802, https://doi.org/10.5194/amt-6-2777-2013, 2013.
- Byrne, B., Liu, J., Bowman, K. W., Pascolini-Campbell, M., Chatterjee, A., Pandey, S., Miyazaki, K., van der Werf, G. R., Wunch, D., Wennberg, P. O., et al.: Carbon emissions from the 2023 Canadian wildfires, Nature, pp. 1–5, https://doi.org/https://doi.org/10.1038/s41586-024-07878-z, 2024.
  - Cao, T., Li, M., Zou, C., Fan, X., Song, J., Jia, W., Yu, C., Yu, Z., and Peng, P.: Chemical composition, optical properties, and oxidative potential of water- and methanol-soluble organic compounds emitted from the combustion of biomass materials and coal, Atmospheric Chemistry and Physics, 21, 13 187–13 205, https://doi.org/10.5194/acp-21-13187-2021, 2021.
- 600 Carrico, C. M., Petters, M. D., Kreidenweis, S. M., Sullivan, A. P., McMeeking, G. R., Levin, E. J. T., Engling, G., Malm, W. C., and Collett Jr., J. L.: Water uptake and chemical composition of fresh aerosols generated in open burning of biomass, Atmospheric Chemistry and Physics, 10, 5165–5178, https://doi.org/10.5194/acp-10-5165-2010, 2010.
  - Carslaw, K. S.: Aerosols and Climate, Elsevier Science Publishing, 1 edn., 2022.
- Czech, H., Popovicheva, O., Chernov, D. G., Kozlov, A., Schneider, E., Shmargunov, V. P., Sueur, M., Rüger, C. P., Afonso, C., Uzhegov, V., et al.: Wildfire plume ageing in the photochemical large aerosol chamber (PHOTO-LAC), Environmental Science: Processes & Impacts, 26, 35–55, https://doi.org/10.1039/D3EM00280B, 2024.
  - Danielsen, E. F.: Stratospheric-tropospheric exchange based on radioactivity, ozone and potential vorticity, Journal of Atmospheric Sciences, 25, 502–518, 1968.
- Dawson, K. W., Ferrare, R. A., Moore, R. H., Clayton, M. B., Thorsen, T. J., and Eloranta, E. W.: Ambient Aerosol Hygroscopic Growth From Combined Raman Lidar and HSRL, Journal of Geophysical Research: Atmospheres, 125, e2019JD031708, https://doi.org/https://doi.org/10.1029/2019JD031708, e2019JD031708 2019JD031708, 2020.
  - Engelhart, G. J., Hennigan, C. J., Miracolo, M. A., Robinson, A. L., and Pandis, S. N.: Cloud condensation nuclei activity of fresh primary and aged biomass burning aerosol, Atmospheric Chemistry and Physics, 12, 7285–7293, https://doi.org/10.5194/acp-12-7285-2012, 2012.
- Garra, P., Maschowski, C., Liaud, C., Dieterlen, A., Trouvé, G., Le Calvé, S., Jaffrezo, J.-L., Leyssens, G., Schönnenbeck, C., Kohler,
   S., and Gieré, R.: Fluorescence microscopy analysis of particulate matter from biomass burning: Polyaromatic Hydrocarbons as Main Contributors, Aerosol Science and Technology, 49, 1160–1169, https://doi.org/10.1080/02786826.2015.1107181, 2015.
  - Gast, B., Jimenez, C., Ansmann, A., Haarig, M., Engelmann, R., Fritzsch, F., Floutsi, A. A., Griesche, H., Ohneiser, K., Hofer, J., Radenz, M., Baars, H., Seifert, P., and Wandinger, U.: Invisible aerosol layers: improved lidar detection capabilities by means of laser-induced aerosol fluorescence, Atmospheric Chemistry and Physics, 25, 3995–4011, https://doi.org/10.5194/acp-25-3995-2025, 2025.
- Haarig, M., Ansmann, A., Baars, H., Jimenez, C., Veselovskii, I., Engelmann, R., and Althausen, D.: Depolarization and lidar ratios at 355, 532, and 1064 nm and microphysical properties of aged tropospheric and stratospheric Canadian wildfire smoke, Atmospheric Chemistry and Physics, 18, 11 847–11 861, https://doi.org/10.5194/acp-18-11847-2018, 2018.
  - Han, S., Hong, J., Luo, Q., Xu, H., Tan, H., Wang, Q., Tao, J., Zhou, Y., Peng, L., He, Y., Shi, J., Ma, N., Cheng, Y., and Su, H.: Hygroscopicity of organic compounds as a function of organic functionality, water solubility, molecular weight, and oxidation level, Atmospheric Chemistry and Physics, 22, 3985–4004, https://doi.org/10.5194/acp-22-3985-2022, 2022.

- Hodshire, A. L., Ramnarine, E., Akherati, A., Alvarado, M. L., Farmer, D. K., Jathar, S. H., Kreidenweis, S. M., Lonsdale, C. R., Onasch, T. B., Springston, S. R., et al.: Dilution impacts on smoke aging: evidence in Biomass Burning Observation Project (BBOP) data, Atmospheric Chemistry and Physics, 21, 6839–6855, https://doi.org/10.5194/acp-21-6839-2021, 2021.
- Hu, Q.: Advanced aerosol characterization using sun/sky photometer and multi-wavelength Mie-Raman lidar measurements, Ph.D. thesis,
  Lille 1, www.theses.fr/2018LILUR078/document, 2018.
  - Hu, Q., Goloub, P., Veselovskii, I., Bravo-Aranda, J.-A., Popovici, I. E., Podvin, T., Haeffelin, M., Lopatin, A., Dubovik, O., Pietras, C., et al.: Long-range-transported Canadian smoke plumes in the lower stratosphere over northern France, Atmospheric Chemistry and Physics, 19, 1173–1193, https://doi.org/10.5194/acp-19-1173-2019, 2019.
- Hu, Q., Goloub, P., Veselovskii, I., and Podvin, T.: The characterization of long-range transported North American biomass burning plumes:
   what can a multi-wavelength Mie–Raman-polarization-fluorescence lidar provide?, Atmospheric Chemistry and Physics, 22, 5399–5414,
   https://doi.org/10.5194/acp-22-5399-2022, 2022.
  - Jain, P., Barber, Q. E., Taylor, S., Whitman, E., Acuna, D. C., Boulanger, Y., Chavardès, R. D., Chen, J., Englefield, P., Flannigan, M., et al.: Canada Under Fire–Drivers and Impacts of the Record-Breaking 2023 Wildfire Season, Authorea Preprints, https://doi.org/https://doi.org/10.22541/essoar.170914412.27504349/v1, 2024.
- Jimenez, J. L., Canagaratna, M. R., Donahue, N. M., Prevot, A. S. H., Zhang, Q., Kroll, J. H., DeCarlo, P. F., Allan, J. D., Coe, H., Ng, N. L., Aiken, A. C., Docherty, K. S., Ulbrich, I. M., Grieshop, A. P., Robinson, A. L., Duplissy, J., Smith, J. D., Wilson, K. R., Lanz, V. A., Hueglin, C., Sun, Y. L., Tian, J., Laaksonen, A., Raatikainen, T., Rautiainen, J., Vaattovaara, P., Ehn, M., Kulmala, M., Tomlinson, J. M., Collins, D. R., Cubison, M. J., E., Dunlea, J., Huffman, J. A., Onasch, T. B., Alfarra, M. R., Williams, P. I., Bower, K., Kondo, Y., Schneider, J., Drewnick, F., Borrmann, S., Weimer, S., Demerjian, K., Salcedo, D., Cottrell, L., Griffin, R., Takami, A., Miyoshi, T.,
- Hatakeyama, S., Shimono, A., Sun, J. Y., Zhang, Y. M., Dzepina, K., Kimmel, J. R., Sueper, D., Jayne, J. T., Herndon, S. C., Trimborn, A. M., Williams, L. R., Wood, E. C., Middlebrook, A. M., Kolb, C. E., Baltensperger, U., and Worsnop, D. R.: Evolution of Organic Aerosols in the Atmosphere, Science, 326, 1525–1529, https://doi.org/10.1126/science.1180353, 2009.
  - June, N. A., Hodshire, A. L., Wiggins, E. B., Winstead, E. L., Robinson, C. E., Thornhill, K. L., Sanchez, K. J., Moore, R. H., Pagonis, D., Guo, H., Campuzano-Jost, P., Jimenez, J. L., Coggon, M. M., Dean-Day, J. M., Bui, T. P., Peischl, J., Yokelson, R. J., Alvarado,
- M. J., Kreidenweis, S. M., Jathar, S. H., and Pierce, J. R.: Aerosol size distribution changes in FIREX-AQ biomass burning plumes: the impact of plume concentration on coagulation and OA condensation/evaporation, Atmospheric Chemistry and Physics, 22, 12 803–12 825, https://doi.org/10.5194/acp-22-12803-2022, 2022.
- Katich, J. M., Apel, E. C., Bourgeois, I., Brock, C. A., Bui, T. P., Campuzano-Jost, P., Commane, R., Daube, B., Dollner, M., Fromm, M., Froyd, K. D., Hills, A. J., Hornbrook, R. S., Jimenez, J. L., Kupc, A., Lamb, K. D., McKain, K., Moore, F., Murphy, D. M., Nault,
  B. A., Peischl, J., Perring, A. E., Peterson, D. A., Ray, E. A., Rosenlof, K. H., Ryerson, T., Schill, G. P., Schroder, J. C., Weinzierl, B., Thompson, C., Williamson, C. J., Wofsy, S. C., Yu, P., and Schwarz, J. P.: Pyrocumulonimbus affect average stratospheric aerosol composition, Science, 379, 815–820, https://doi.org/10.1126/science.add3101, 2023.
  - Khaykin, S., Bekki, S., Godin-Beekmann, S., Fromm, M. D., Goloub, P., Hu, Q., Josse, B., Laeng, A., Meziane, M., Peterson, D. A., Pelletier, S., and Thouret, V.: Stratospheric impact of the anomalous 2023 Canadian wildfires: the two vertical pathways of smoke, EGUsphere, 2025, 1–27, https://doi.org/10.5194/egusphere-2025-3152, 2025.
  - Kleinman, L. I., Sedlacek III, A. J., Adachi, K., Buseck, P. R., Collier, S., Dubey, M. K., Hodshire, A. L., Lewis, E., Onasch, T. B., Pierce, J. R., et al.: Rapid evolution of aerosol particles and their optical properties downwind of wildfires in the western US, Atmospheric Chemistry and Physics, 20, 13 319–13 341, https://doi.org/10.5194/acp-20-13319-2020, 2020.

680

- Krüger, K., Schäfler, A., Wirth, M., Weissmann, M., and Craig, G. C.: Vertical structure of the lower-stratospheric moist bias in the ERA5 reanalysis and its connection to mixing processes, Atmospheric Chemistry and Physics, 22, 15559–15577, https://doi.org/10.5194/acp-22-15559-2022, 2022.
  - Kuang, Y., Xu, W., Tao, J., Ma, N., Zhao, C., and Shao, M.: A review on laboratory studies and field measurements of atmospheric organic aerosol hygroscopicity and its parameterization based on oxidation levels, Current Pollution Reports, 6, 410–424, 2020.
- Lambe, A. T., Onasch, T. B., Massoli, P., Croasdale, D. R., Wright, J. P., Ahern, A. T., Williams, L. R., Worsnop, D. R., Brune, W. H., and Davidovits, P.: Laboratory studies of the chemical composition and cloud condensation nuclei (CCN) activity of secondary organic aerosol (SOA) and oxidized primary organic aerosol (OPOA), Atmospheric Chemistry and Physics, 11, 8913–8928, https://doi.org/10.5194/acp-11-8913-2011, 2011.
  - Lee, H. J., Laskin, A., Laskin, J., and Nizkorodov, S. A.: Excitation–emission spectra and fluorescence quantum yields for fresh and aged biogenic secondary organic aerosols, Environmental science & technology, 47, 5763–5770, https://doi.org/10.1021/es400644c, 2013.
- Mamouri, R.-E., Ansmann, A., Ohneiser, K., Knopf, D. A., Nisantzi, A., Bühl, J., Engelmann, R., Skupin, A., Seifert, P., Baars, H., Ene, D., Wandinger, U., and Hadjimitsis, D.: Wildfire smoke triggers cirrus formation: lidar observations over the eastern Mediterranean, Atmospheric Chemistry and Physics, 23, 14 097–14 114, https://doi.org/10.5194/acp-23-14097-2023, 2023.
  - Massoli, P., Lambe, A. T., Ahern, A. T., Williams, L. R., Ehn, M., Mikkilä, J., Canagaratna, M. R., Brune, W. H., Onasch, T. B., Jayne, J. T., Petäjä, T., Kulmala, M., Laaksonen, A., Kolb, C. E., Davidovits, P., and Worsnop, D. R.: Relationship between aerosol oxidation level and hygroscopic properties of laboratory generated secondary organic aerosol (SOA) particles, Geophysical Research Letters, 37, https://doi.org/https://doi.org/10.1029/2010GL045258, 2010.
  - Müller, D., Mattis, I., Ansmann, A., Wandinger, U., Ritter, C., and Kaiser, D.: Multiwavelength Raman lidar observations of particle growth during long-range transport of forest-fire smoke in the free troposphere, Geophysical Research Letters, 34, https://doi.org/10.1029/2006GL027936, 2007.
- Navas-Guzmán, F., Martucci, G., Collaud Coen, M., Granados-Muñoz, M. J., Hervo, M., Sicard, M., and Haefele, A.: Characterization of aerosol hygroscopicity using Raman lidar measurements at the EARLINET station of Payerne, Atmospheric Chemistry and Physics, 19, 11 651–11 668, https://doi.org/10.5194/acp-19-11651-2019, 2019.
  - Ohneiser, K., Ansmann, A., Baars, H., Seifert, P., Barja, B., Jimenez, C., Radenz, M., Teisseire, A., Floutsi, A., Haarig, M., et al.: Smoke of extreme Australian bushfires observed in the stratosphere over Punta Arenas, Chile, in January 2020: optical thickness, lidar ratios, and depolarization ratios at 355 and 532 nm, Atmospheric Chemistry and Physics, 20, 8003–8015, https://doi.org/10.5194/acp-20-8003-2020, 2020.
    - Ohneiser, K., Ansmann, A., Kaifler, B., Chudnovsky, A., Barja, B., Knopf, D. A., Kaifler, N., Baars, H., Seifert, P., Villanueva, D., et al.: Australian wildfire smoke in the stratosphere: the decay phase in 2020/2021 and impact on ozone depletion, Atmospheric Chemistry and Physics, 22, 7417–7442, https://doi.org/https://doi.org/10.5194/acp-22-7417-2022, 2022.
- Perring, A. E., Schwarz, J. P., Markovic, M. Z., Fahey, D. W., Jimenez, J. L., Campuzano-Jost, P., Palm, B. D., Wisthaler, A., Mikoviny, T., Diskin, G., Sachse, G., Ziemba, L., Anderson, B., Shingler, T., Crosbie, E., Sorooshian, A., Yokelson, R., and Gao, R.-S.: In situ measurements of water uptake by black carbon-containing aerosol in wildfire plumes, Journal of Geophysical Research: Atmospheres, 122, 1086–1097, https://doi.org/10.1002/2016JD025688, 2017.
- Peterson, D. A., Fromm, M. D., Solbrig, J. E., Hyer, E. J., Surratt, M. L., and Campbell, J. R.: Detection and Inventory of Intense Pyroconvection in Western North America using GOES-15 Daytime Infrared Data, Journal of Applied Meteorology and Climatology, 56, 471 493, https://doi.org/10.1175/JAMC-D-16-0226.1, 2017.

- Petters, M. D., Carrico, C. M., Kreidenweis, S. M., Prenni, A. J., DeMott, P. J., Collett Jr, J. L., and Moosmüller, H.: Cloud condensation nucleation activity of biomass burning aerosol, Journal of Geophysical Research: Atmospheres, 114, 2009.
- Pöhlker, M. L., Pöhlker, C., Quaas, J., Mülmenstädt, J., Pozzer, A., Andreae, M. O., Artaxo, P., Block, K., Coe, H., Ervens, B., et al.: Global organic and inorganic aerosol hygroscopicity and its effect on radiative forcing, Nature communications, 14, 6139, 2023.
  - Reichardt, J.: Cloud and Aerosol Spectroscopy with Raman Lidar, Journal of Atmospheric and Oceanic Technology, 31, 1946 1963, https://doi.org/10.1175/JTECH-D-13-00188.1, 2014.
  - Reichardt, J., Lauermann, F., and Behrendt, O.: Aerosol Studies with Spectrometric Fluorescence and Raman Lidar, in: International Laser Radar Conference, pp. 279–285, Springer, https://doi.org/https://doi.org/10.1007/978-3-031-37818-8\_37, 2022.
- Reichardt, J., Behrendt, O., and Lauermann, F.: Spectrometric fluorescence and Raman lidar: absolute calibration of aerosol fluorescence spectra and fluorescence correction of humidity measurements, Atmospheric Measurement Techniques, 16, 1–13, https://doi.org/https://doi.org/10.5194/amt-16-1-2023, 2023.
  - Reichardt, J., Lauermann, F., and Behrendt, O.: Fluorescence spectra of atmospheric aerosols, Atmospheric Chemistry and Physics, 25, 5857–5892, https://doi.org/10.5194/acp-25-5857-2025, 2025.
- Rosenfeld, D., Fromm, M., Trentmann, J., Luderer, G., Andreae, M. O., and Servranckx, R.: The Chisholm firestorm: observed microstructure, precipitation and lightning activity of a pyro-cumulonimbus, Atmospheric Chemistry and Physics, 7, 645–659, https://doi.org/10.5194/acp-7-645-2007, 2007.
  - Rudich, Y., Donahue, N. M., and Mentel, T. F.: Aging of organic aerosol: Bridging the gap between laboratory and field studies, Annu. Rev. Phys. Chem., 58, 321–352, 2007.
- Schill, G., Froyd, K., Bian, H., Kupc, A., Williamson, C., Brock, C., Ray, E., Hornbrook, R., Hills, A., Apel, E., et al.: Widespread biomass burning smoke throughout the remote troposphere, Nature Geoscience, 13, 422–427, 2020.
  - Selimovic, V., Yokelson, R. J., McMeeking, G. R., and Coefield, S.: In situ measurements of trace gases, PM, and aerosol optical properties during the 2017 NW US wildfire smoke event, Atmospheric Chemistry and Physics, 19, 3905–3926, https://doi.org/10.5194/acp-19-3905-2019, 2019.
- Simmons, A., Soci, C., Nicolas, J., Bell, B., Berrisford, P., Dragani, R., Flemming, J., Haimberger, L., Healy, S., Hersbach, H., et al.: Global stratospheric temperature bias and other stratospheric aspects of ERA5 and ERA5. 1, European Centre for Medium Range Weather Forecasts Reading, UK, 146, 1951–1972, https://doi.org/10.1002/qj.3803, 2020.
  - Solomon, S., Dube, K., Stone, K., Yu, P., Kinnison, D., Toon, O. B., Strahan, S. E., Rosenlof, K. H., Portmann, R., Davis, S., Randel, W., Bernath, P., Boone, C., Bardeen, C. G., Bourassa, A., Zawada, D., and Degenstein, D.: On the stratospheric chemistry of midlatitude wildfire smoke, Proceedings of the National Academy of Sciences, 119, e2117325119, https://doi.org/10.1073/pnas.2117325119, 2022.
  - Sugimoto, N., Huang, Z., Nishizawa, T., Matsui, I., and Tatarov, B.: Fluorescence from atmospheric aerosols observed with a multi-channel lidar spectrometer, Opt. Express, 20, 20 800–20 807, https://doi.org/10.1364/OE.20.020800, 2012.
  - Sun, B., Calbet, X., Reale, A., Schroeder, S., Bali, M., Smith, R., and Pettey, M.: Accuracy of Vaisala RS41 and RS92 Upper Tropospheric Humidity Compared to Satellite Hyperspectral Infrared Measurements, Remote Sensing, 13, https://doi.org/10.3390/rs13020173, 2021.
- 735 Trickl, T., Vogelmann, H., Flentje, H., and Ries, L.: Stratospheric ozone in boreal fire plumes–the 2013 smoke season over central Europe, Atmospheric Chemistry and Physics, 15, 9631–9649, 2015.
  - Veselovskii, I., Hu, Q., Goloub, P., Podvin, T., Korenskiy, M., Pujol, O., Dubovik, O., and Lopatin, A.: Combined use of Mie–Raman and fluorescence lidar observations for improving aerosol characterization: feasibility experiment, Atmospheric Measurement Techniques, 13, 6691–6701, https://doi.org/10.5194/amt-13-6691-2020, 2020.

- 740 Veselovskii, I., Kasianik, N., Korenskii, M., Hu, Q., Goloub, P., Podvin, T., and Liu, D.: Multiwavelength fluorescence lidar observations of smoke plumes, Atmospheric Measurement Techniques, 16, 2055–2065, https://doi.org/10.5194/amt-16-2055-2023, 2023.
  - Veselovskii, I., Hu, Q., Goloub, P., Podvin, T., Boissiere, W., Korenskiy, M., Kasianik, N., Khaykyn, S., and Miri, R.: Derivation of depolarization ratios of aerosol fluorescence and water vapor Raman backscatters from lidar measurements, Atmospheric Measurement Techniques, 17, 1023–1036, https://doi.org/10.5194/amt-17-1023-2024, 2024a.
- Veselovskii, I., Korenskiy, M., Kasianik, N., Barchunov, B., Hu, Q., Goloub, P., and Podvin, T.: Fluorescence properties of long-range transported smoke: Insights from five-channel lidar observations over Moscow during the 2023 wildfire season, EGUsphere, 2024, 1–23, https://doi.org/10.5194/egusphere-2024-2874, 2024b.
  - Veselovskii, I., Korenskiy, M., Kasianik, N., Barchunov, B., Hu, Q., Goloub, P., and Podvin, T.: Fluorescence properties of long-range-transported smoke: insights from five-channel lidar observations over Moscow during the 2023 wildfire season, Atmospheric Chemistry and Physics, 25, 1603–1615, https://doi.org/10.5194/acp-25-1603-2025, 2025.
  - Wallace, J. M. and Hobbs, P. V.: Atmospheric science: an introductory survey, vol. 92, Elsevier, 2006.
  - Wang, J., Shilling, J. E., Liu, J., Zelenyuk, A., Bell, D. M., Petters, M. D., Thalman, R., Mei, F., Zaveri, R. A., and Zheng, G.: Cloud droplet activation of secondary organic aerosol is mainly controlled by molecular weight, not water solubility, Atmospheric Chemistry and Physics, 19, 941–954, https://doi.org/10.5194/acp-19-941-2019, 2019.
- Yu, P., Toon, O. B., Bardeen, C. G., Zhu, Y., Rosenlof, K. H., Portmann, R. W., Thornberry, T. D., Gao, R.-S., Davis, S. M., Wolf, E. T., de Gouw, J., Peterson, D. A., Fromm, M. D., and Robock, A.: Black carbon lofts wildfire smoke high into the stratosphere to form a persistent plume, Science, 365, 587–590, https://doi.org/10.1126/science.aax1748, 2019.
  - Zhang, Y., Wang, L., Liu, P., Li, Y., Zhan, R., Huang, Z., and Lin, H.: Measurement and extrapolation modeling of PAH laser-induced fluorescence spectra at elevated temperatures, Applied Physics B, 125, 1–12, https://doi.org/10.1007/s00340-018-7115-6, 2019.
- Zheng, G., Sedlacek, A. J., Aiken, A. C., Feng, Y., Watson, T. B., Raveh-Rubin, S., Uin, J., Lewis, E. R., and Wang, J.: Long-range transported North American wildfire aerosols observed in marine boundary layer of eastern North Atlantic, Environment International, 139, 105 680, https://doi.org/https://doi.org/10.1016/j.envint.2020.105680, 2020.