# Peer review of "Advanced insights into biomass burning aerosols during the 2023 Canadian wildfires from dual-site Raman and fluorescence lidar observations"

_EGUsphere, 2025_

## Referee Comment (RC2)

**Comments on "Advanced insights into biomass burning aerosols during the 2023 Canadian wildfires from dual-site Raman and fluorescence lidar observations" by Qiaoyun Hu et al.**

**Summary:**

This manuscript presents lidar observations of long-range transported biomass burning aerosol (BBA) at two stations: Lille (France) and Moscow (Russia). Firstly, two case studies from each station are discussed to show comparability between the results at both sites. Furthermore, statistical results of the BBA optical properties from five months of observation in 2023 are presented and discussed. The fluorescence capacities are also compared to two additional stations in Germany, showing a general agreement in magnitude for all four lidar stations.

**General comment:**

The results presented are of high quality and this study advances the knowledge on optical properties of long-range transported BBA. The general agreement of laser-induced fluorescence observations at different stations suggests a potential for coordinated multi-lidar fluorescence measurements in the future. Therefore, this study is an important contribution to aerosol research and well-suited for ACP.

In general, the manuscript is well written, I would only suggest minor revisions regarding the following aspects:

- Style of the figures: In some figures, the tick labels of the axes are rather small and some thin colored lines are difficult to distinguish for colorblind people. Please increase the label size and choose more contrasting colors, where necessary, to improve readability. For details, please refer to the specific comments below.
- Fluorescence channel calibration (Appendix A): By exchanging the PMTs of the nitrogen Raman and fluorescence channel, only the difference in electrical gain between the two PMTs is obtained, as Eq. A3 is building the ratio between both PMTs at the same wavelength (387 nm in your case). However, PMTs do also show a spectral dependence of the sensitivity, which will influence the ratio of the PMT efficiencies ($g_R/g_F$). Did you check the spectral dependence of the PMTs (e.g., with information from the manufacturer) used and account for it? Two further questions are addressed in the specific comments.

**Specific comments:**

**l. 46:** Ansmann et al. (2025a) also discussed the impact of BBA on cirrus cloud formation in the Arctic. This reference should be added here.

**ll. 54-55:** I would recommend to sort the references to the two types of LIF lidar systems. Thus, it would be easier to understand for readers that are new to the technique.

**l. 102:** Should it be "opt**o**-electronic" here?

**l. 130:** Eq. 4: Why did you select the ratio of exactly these two channels for the color ratio? Please explain.

**l. 146:** Are there really mixed-phase clouds present? The depolarization ratio seems high throughout the whole cloud area, indicating non-spherical ice crystals.

**Figure 2:** Please take care of the units in the colormap labels. I assume, the Backscatter coefficient in panel (a) should be in m$^{-1}$ sr$^{-1}$ (so it is also stated in the figure caption). Panel (b) is missing the unit at all. The same comment holds for Fig. 4.

**ll. 197-198:** "Giant and dense smoke plumes  can be observed from MODIS observations in Figure B1(b)."
This is not a correct sentence. I would suggest to omit the "which". Furthermore, case 2 is shown in Fig. B1(**a**).

**ll. 199-200:** "One notable feature is that BBA layers at 12000 m showed higher depolarization ratio at 532 nm and fluorescence capacity than those in the free troposphere, as shown in Figure 4(b) and (d)".
There is no fluorescence capacity s hown in Fig. 4. Panel (d) is obviously missing in Fig. 4.

**ll. 222-223:** Besides the stronger fluorescence signal, the second plume also showed a higher depolarization ratio at 355 nm, or not? Please add that.

**Figure 6:** Shouldn't the unit of the spectral fluorescence backscatter coefficient be something like Mm$^{-1}$ sr$^{-1}$ **nm$^{-1}$** (per nm)? The same comment applies to Fig. 8.

**Figure 7:** Panels (d) and (e): The two different blue tones are difficult to distinguish for colorblind people. Please change one of the tones to a more contrasting one. Maybe use the same combination blue tones as for the RH from sonde and ERA 5 data. They are well distinguishable. Thank you. The same comment applies to Fig. 9.

**l. 279:** *"The increasing trend is particularly strong in the thin BBA layer at 7000 to 9000 m."*
You only show the color ratio up to 8 km but mention, that the trend is strong from 7-9 km. What happens from 8-9 km?

**Figure 10:** Please increase the size of the tick labels of the axes for better readability.

**ll. 354-355:** *"Exceptionally high depolarization ratios were observed during two periods: on 12 July and from 23 to 30 September 2023"*
And at the end of August, if I'm right?

**ll. 360-361:** *". In particular, LILAS detected tropospheric BBA layers with depolarization ratios comparable to those in the UTLS on 29 and 30 September"*
There seems to be a similar behavior at the end of August, isn't it?

**l. 381:** *"However, in the UTLS layer MARTHA derived much lower values than LILAS."*
You only compare MARTHA and LILAS here. If you would compare the RAMSES and GPI data to them as well, this could be also interesting: Then, it seems that MARTHA and RAMSES values are nearer to each other than LILAS and MARTHA (which seems reasonable, as Lindenberg and Leipzig are closer to each other than Lille and Leipzig). LILAS, instead is quite near to T2 values of GPI, while RAMSES and MARTHA are closer to T1 of GPI.

**l. 389:** *"over DWD"* → "over **Lindenberg**" would be better here.

**Table C1:** The numbers for the layers around 12 km from LILAS and MARTHA are confusing (compared to Fig. 12, they are swapped). In Fig. 12, LILAS shows higher capacity, while in Tab. C1, MARTHA shows a higher value than LILAS. Please clarify.

**Figure 13:** What is represented by the dashed lines – the average for FT and UTLS BBA layers, respectively? This would not match with the values reported in the text. Please clarify.

**ll. 429-430:** *"The EAE decreases gradually with the increasing layer altitude, yielding r2 ≈ 0.61, while the depolarization ratio exhibits an increasing trend with r2 ≈0.68."*

You only show the depolarization ratio at 532 nm in Fig. 15b. Was it the same at 355 nm or did the depolarization ratio show no tendency there?

**Figure 15:** Please increase the size of the axes and tick labels for better readability.

**ll. 553-554:** *"Note that, in Equation A1-A3, we assume the transmission in the optics before the DM is invariant between the spectral range of the Raman and fluorescence channel."* Is that really the case or an approximative assumption? Are there no other optical elements (such as other beam splitters or dichroic mirrors) in the beam path before, that may have different transmissions for the Raman and fluorescence wavelength ranges?

**Figure B1:** I would recommend to switch the order of the two graphics, here. Because at the moment, panel (a) shows case 2 and panel (b) shows case 1, which is bit confusing. After switching, it would be more chronological.

**Typos:**

**l. 241:** *"wavelengths"*

**l. 266:** "*HYPLIT*" → HY**S**PLIT

**l. 419:** *"clearly"* → clear; *"toward"* → toward**s**

**l. 424:** *"upper most"* → uppermost

**l. 540:** *"interferences filter"* → interference filter**s**

**l. 555:** *"grain"* → gain

**References:**

Ansmann, A., Jimenez, C., Roschke, J., Bühl, J., Ohneiser, K., Engelmann, R., Radenz, M., Griesche, H., Hofer, J., Althausen, D., Knopf, D. A., Dahlke, S., Gaudek, T., Seifert, P., and Wandinger, U.: Impact of wildfire smoke on Arctic cirrus formation – Part 1: Analysis of MOSAiC 2019–2020 observations, Atmos. Chem. Phys., 25, 4847–4866, https://doi.org/10.5194/acp-25-4847-2025, 2025.